# Responses of Ground-Dwelling Spider (Arachnida: Araneae) Communities to Wildfire in Three Habitats in Northern New Mexico, USA, with Notes on Mites and Harvestmen (Arachnida: Acari, Opiliones)

**Sandra L. Brantley**

Museum of Southwestern Biology, University of New Mexico, Albuquerque, NM 87131, USA;
sbrantle@unm.edu; Tel.: +1-505-620-6838

**Abstract:** Catastrophic wildfire is increasingly common in forests of the western United States because climate change is increasing ambient temperatures and periods of drought. In 2011, the Las Conchas wildfire burned in the Santa Fe National Forest of New Mexico, including portions of ponderosa pine and mixed-conifer forests, and grasslands in the Valles Caldera National Preserve, a large, high-elevation volcanic caldera. Following the fire, Caldera staff began monitoring abiotic, plant, and animal responses. In this study, ground-dwelling arachnids were collected in pitfall traps in burned and unburned habitats from 2011–2015. Permutational multivariate analysis of variance (PERMANOVA) mostly at the genus level with some higher taxon levels showed significant fire, year, and interaction effects. Abundance was at or near unburned levels by 2014, but species composition changed in burned areas. *Pardosa* and *Haplodrassus* were dominant genera across habitats. Linyphiids were strong indicators of unburned sites. Harvestmen were among the dominant species in the forest habitats, and erythraeid mites were abundant in the burned ponderosa pine forest and the grassland. Years were not significantly autocorrelated, unsurprising given the interannual variation in precipitation in this generally arid region. Although fire is a common feature of these habitats, future fires may be outside of historical patterns, preventing spider communities from re-establishing fully.

**Keywords:** Araneae; Acari; Opiliones; wildfire; forests; New Mexico; pitfall trapping

## 1. Introduction

In 2011, a drought year, the largest wildfire (63,371 ha) in New Mexico, USA up to that time burned in the Santa Fe National Forest, from 26 June to 01 August [1]. The catastrophic Las Conchas fire was one more example of increasing wildfires in the western region of the U.S., as temperatures and drought periods have been increasing [2–4]. In the southern Rocky Mountains, several climate change factors are influencing the frequency and severity of fires in this region: trends toward larger fires, warmer maximum air temperatures between September and November, less precipitation between June and November, increased drought severity [2]. Results from a study of the years 1973–2012 in the western United States suggested trends of overall lengthening of the fire season from 37–117 days and mean burn time from 5 to 37 days [4]. Early snowmelt from the mountains also increased the likelihood of large summer wildfires [4]. In the Colorado Front Range of the Rocky Mountains, Rother and Veblen [3] looked at stands of ponderosa pine from past fires to estimate what future climate changes could mean for tree establishment. They found that the severity of the burn was less important than the extent of the fire, making the distance to seed sources for new trees longer. More vulnerable stands of ponderosa pine were at their lower elevation level or were found on south-facing slopes. The effects

of fires in forests are complex; season, tree species, and local habitat features (such as slope or aspect) are all important in affecting the outcome of wildfire.

This study took place in the Jemez Mountains at the Valles Caldera National Preserve, which is within the National Forest and which experienced stand-replacing burns in the ponderosa pine forest, moderate to severe burns in the mixed-conifer forest, and less severe burns in mountain grassland in 2011. At the Caldera, monitoring of abiotic factors and biotic responses began as soon as the fire was contained, through the Collaborative Forest Landscape Restoration Project (CFLRP) for the southwest Jemez Mountains. The CFLRP had been established in 2010 to restore forest resilience to wildfire and other disturbances and improve conditions for wildlife, watersheds, and vegetation [5]. Ground-dwelling spiders, harvestmen, and selected mite groups were among the animal taxa chosen to be monitored from 2011 to 2015. They had not previously been surveyed at the Caldera, so this monitoring effort also provided the opportunity to add to the known diversity and distribution of these groups in a region that is relatively understudied (but see [6–8]).

These three arachnid taxa are widespread generalist predators, an important arthropod trophic group, with over 3800 species of spiders alone in North America, and are also a food source for larger animals, particularly birds and reptiles [9,10]. Because they are not herbivores, their response to post-fire changes in vegetation is based less on plant species and more on plant structure and litter amount [9,11]. They are frequent early colonizers to disturbed areas, whether walking in from nearby areas or through dispersal by ballooning [9,10].

There are numerous papers on spider responses to prescribed fires, with and without the combined treatment of timber thinning [12–14], an increasing number on wildfire [6,15–17], but understandably fewer with information on pre-fire conditions for the target species [18]. In different forests, spider abundance was either not affected by prescribed burns or returned to pre-fire levels within 3 years: Oregon, USA [12], in Swedish boreal forest [14], and in the juniper-poplar steppe in Hungary [17]. Wildfire sometimes more strongly affected spider communities. In the forest in Finland [15], spider assemblages were clearly different 3 years post-fire. In Canada spruce forest, a comparison of clear-cutting and wildfire [16] showed that spider responses to the two treatments were different and that wildfire (that is not catastrophic) could leave a more heterogeneous litter layer and thus had a less pronounced habitat effect than clear-cutting. In an Oregon, USA, grass/shrub steppe [18], study sites were in place before a wildfire occurred. There was no difference among sites before the fire, but afterward community composition changed, although, at the broader scale, richness and abundance did not. One study from Colorado, USA [6] looked at the effects of a 2002 catastrophic wildfire in pinyon-juniper woodland. Five to six years later, vegetation cover of grasses and annual plants had increased, litter decreased, and bare ground increased, compared with nearby old-growth stands. None of the 32 spider species was in the top 7 indicators of burned or unburned sites, but abundance and richness had not returned to levels found in old-growth controls. Four spider species were positively associated with the burned areas and 16 were negatively associated with the burned areas. Given the variability in extent and severity of wildfire, it is difficult to compare it to prescribed fire; therefore, additional studies from wildfire are needed. This is especially important for those areas that may be forested but situated in largely arid regions, as is the case for much of the southwestern United States.

For the arachnids in the two forest habitats and grassland that were burned in the Caldera, the main questions of interest for the CFLRP program were: (1) Did the Las Conchas fire have an effect on arachnid activity abundance and species composition from 2011 to 2015? If so, (2) By the end of 2015 were burned areas still different from the unburned areas in activity abundance, and species composition? These questions were important to land managers at the Caldera in helping them decide how long monitoring efforts should be. Wildfires in forests and grasslands in this region have been common historically [2,3,19], but currently are increasing in severity and frequency [20], so learning more about the ability of animal species resistance and resilience to them is necessary. Spiders in

general are in the center of trophic food webs, affecting populations of soil mesofauna, insect larvae, and flying insects, and in turn being affected by other arthropod taxa and vertebrate animals.

## 2. Materials and Methods

The Valles Caldera National Preserve is located in the Jemez Mountains in north-central New Mexico, USA (35°55′12″ N,106°31′15.6″ W), part of the Rio Grande rift valley (Figure 1). A volcanic eruption about 1.25 Mya formed the caldera, which is now about 22 km in diameter [21] and ranges in elevation from about 2550 m to 3230 m at the highest point, Redondo Peak. The main vegetation types are forests of ponderosa pine, mixed-conifer, and spruce-fir, as well as grassy high-elevation meadows. For this study, vegetation types included ponderosa pine forest (PP), mixed-conifer forest (MC), and mountain valley grassland (MV), with elevations between 2590 m and 2750 m. *Pinus ponderosa* dominated the PP habitat, and *Pseudotsuga menziesii* (Douglas-fir) and *Picea englemannii* (Englemann spruce) were dominant at MC. At MV, C3 grasses, such as *Festuca* spp., *Danthonia parryi*, and *Poa pratensis*, along with the C4 grass *Muhlenbergia montana,* dominated [22].

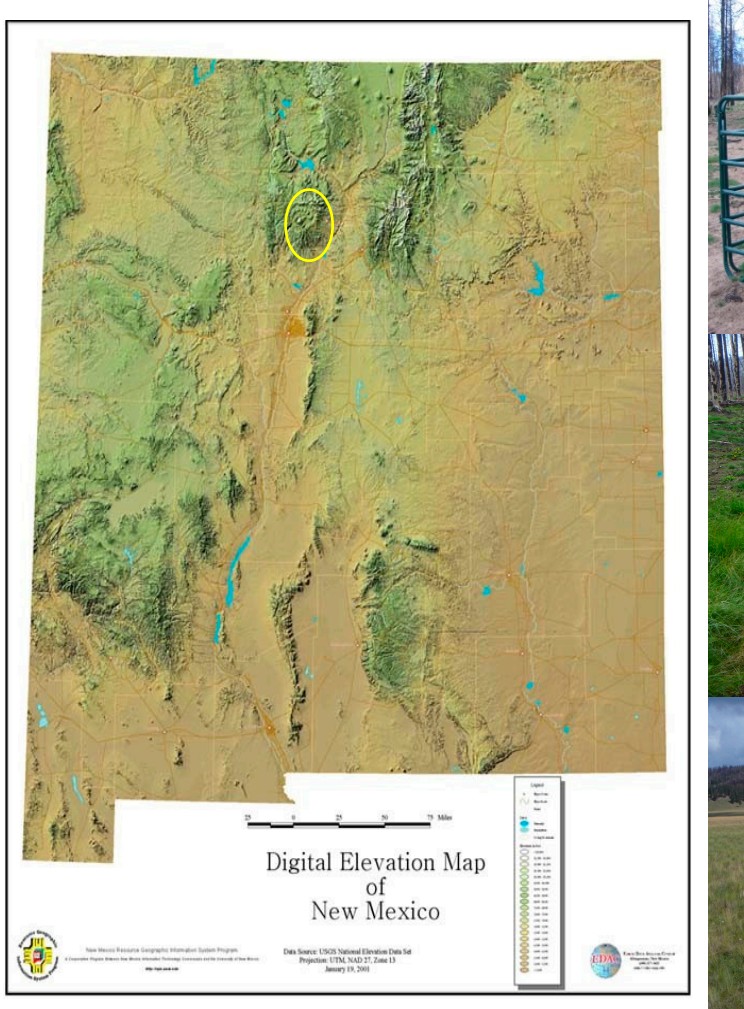
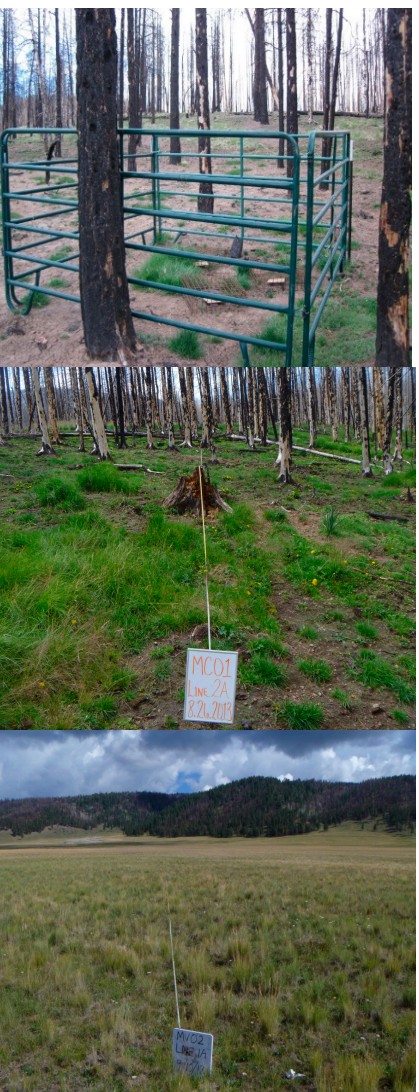

**Figure 1.** Map of New Mexico. Yellow oval outlines the Valles Caldera National Preserve. The three habitats affected by the Las Conchas fire: Ponderosa pine forest, top; Mixed-conifer forest, middle; Mountain Valley Grassland, bottom. The ponderosa pine image shows the pitfall and fencing arrangement.

July maximum air temperatures ranged from 30 to 33 °C and January minimum air temperatures from −23 to −32 °C. Soil temperatures were buffered somewhat for summer temperatures (23–29 °C) and more strongly for winter minimums (−4 to −6 °C). Rainfall occurred mostly during the summer ("monsoon") months [23]. Vegetation cover and exposed litter were regularly monitored by Caldera staff (Figure 2). In New Mexico, most rainfall occurs primarily during July–September (summer or monsoon pattern). At the Caldera from 2011–2015, rainfall ranged from 111 to 160 mm per month in the summer, while amounts ranged from 0 to 50 mm per month at other times of the year. Drought conditions were described by the Palmer Drought Severity Index, which measures departures from normal years in temperature and precipitation. Positive values are wetter years, negative values are dry or drought years. The year 2011 was in moderate to extreme drought (index values of −2.00 to −4.00), 2012 was in severe to extreme drought (−3.00 to −4.00), 2013 was in the mid-range (neither drought nor wet) to severe drought (−1.99 to −3.00), 2014 was in the mid-range to moderate drought (−1.99 to −2.00), and 2015 was in mid-range to very moist conditions (−1.99 to +3.99) [24].

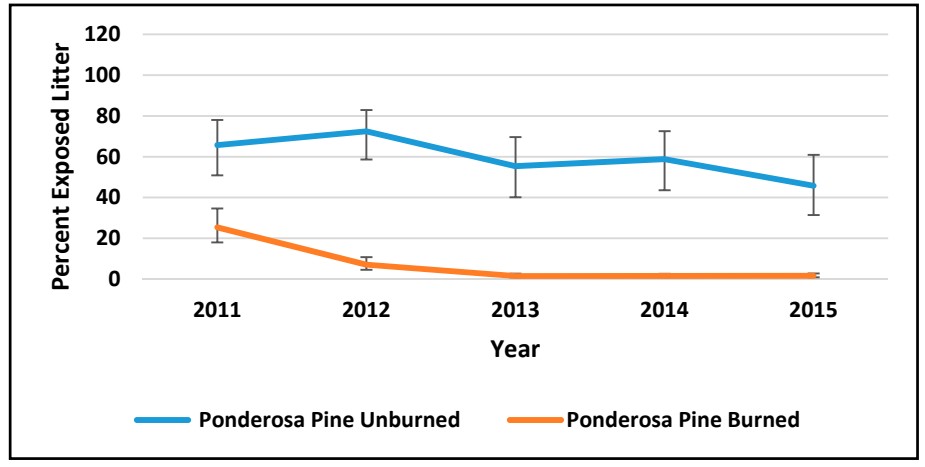

(**a**)

(**b**)

**Figure 2.** *Cont.*

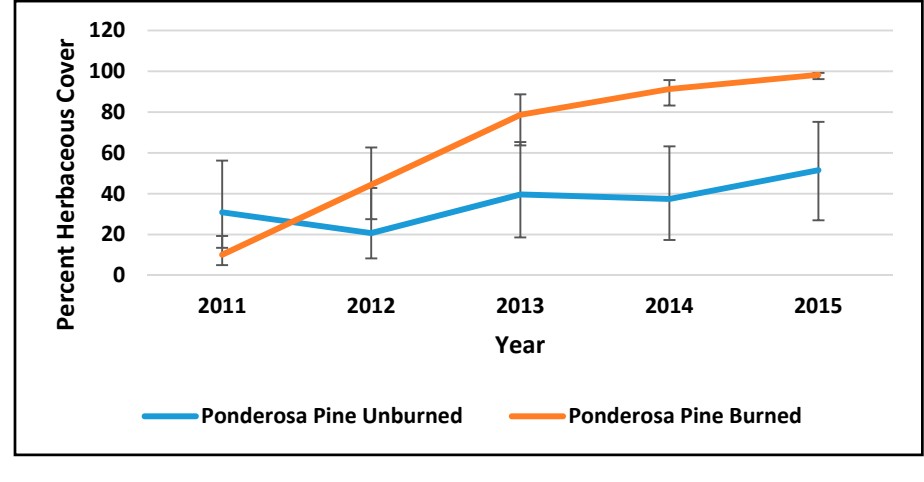

(**c**)

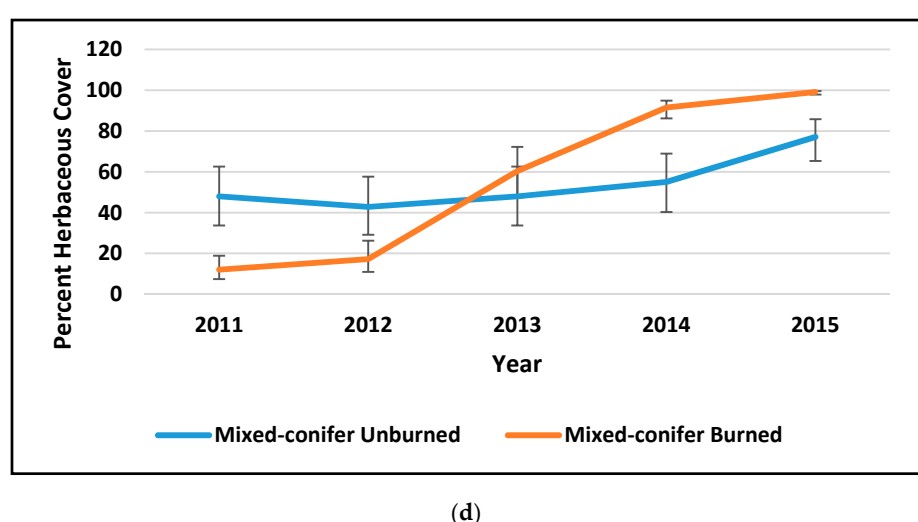

(**d**)

**Figure 2.** Percent cover by year of exposed litter (**a**,**b**) and percent herbaceous cover (**c**,**d**) in ponderosa pine forest (PP) and mixed-conifer forest (MC) at the Valles Caldera (data courtesy Caldera staff). Error bars are 95% confidence intervals. Vegetation cover in Mountain Valley Grassland was always between 90% and 100%; see Suazo et al. (2018) for more detail.

Within each habitat, 12 sites were chosen, 6 in burned areas and 6 in unburned areas (Figure 3). Sites were chosen away from habitat edges and were at least 300–500 m apart, farther if possible, but burn patterns influenced site placement. Three pitfall traps with cups 9 cm in diameter × 12.5 cm in height were placed 1 m apart at each site, each trap about 2/3 filled with propylene glycol as the sample preservative and protected from elk disturbance by fencing (Figure 1). Traps were emptied every two to four weeks between May and early November, except in 2011, when trapping began after the fire, in July 2011. This produced 8–9 sampling periods per year in 2012–2015. In 2013, the Thompson Ridge fire burned 3 of the control sites each at PP and MC, which were removed from the analysis, resulting in an uneven sample set for 2013–2015; the MV sites were not affected by the Thompson Ridge fire.

## Valles Caldera National Preserve

Las Conchas Fire Response Monitoring Sites
Mixed Conifer (MC) and Ponderosa Pine (PP) and Mountain Valley (MV)

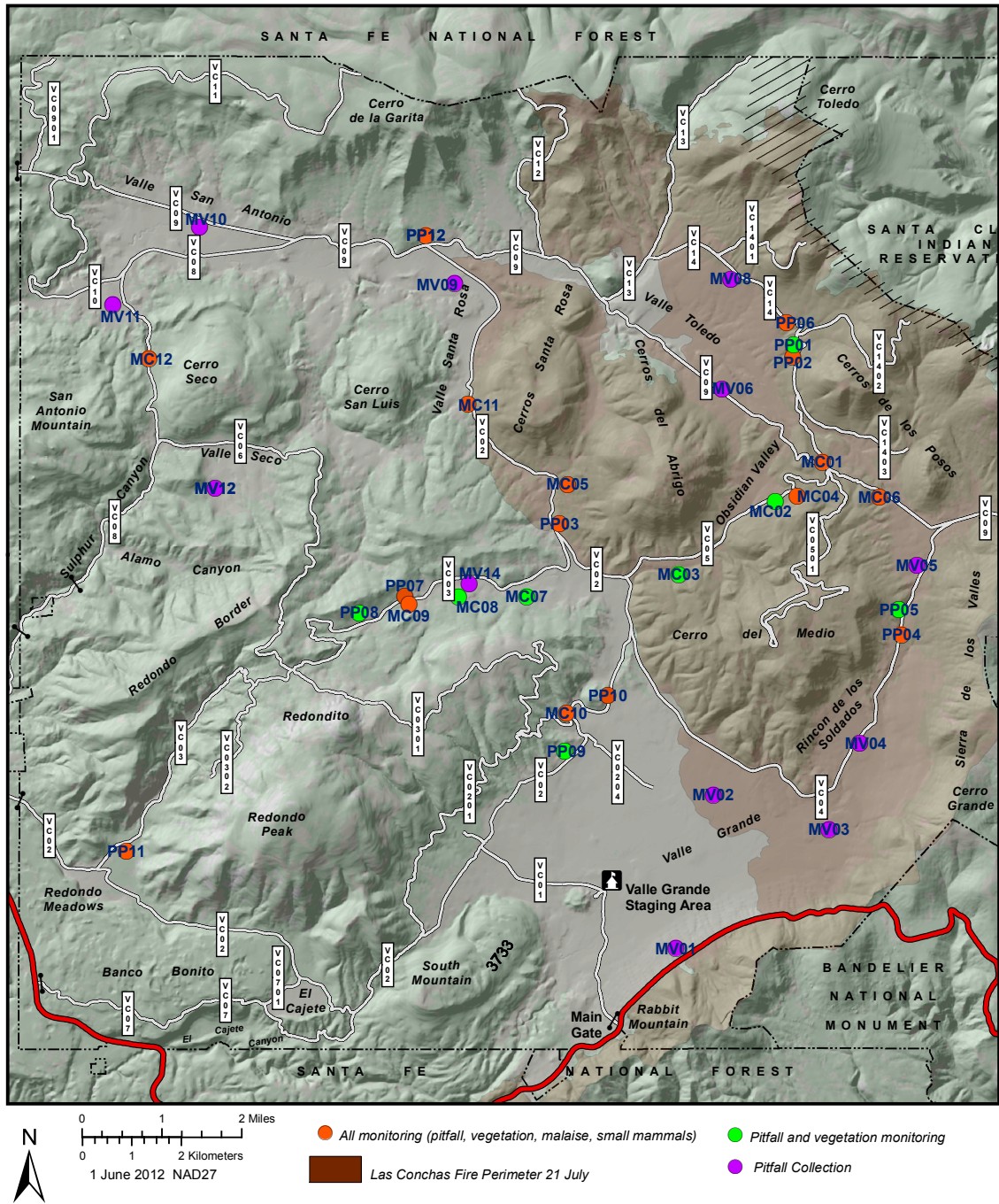

**Figure 3.** Map of the extent of the Las Conchas wildfire study sites for arthropod pitfall traps at the Valles Caldera National Preserve, and the location of pitfall trap sites. Burned areas are to the right; unburned areas are to the left on the map.

Arachnids were separated from other arthropods, identified and counted in the lab at the University of New Mexico Museum of Southwestern Biology. More than 1700 representative specimens, in 70% ethanol, were deposited with the Museum; specimen collection information and catalog numbers were entered into the SCAN (Symbiota Collections of Arthropods Network) arthropod database.

Spiders, harvestmen, and selected mite groups and their abundances were included for analysis (Tables A1–A3); however, if there were fewer than 10 individuals within a habitat and taxon over the course of the study, they were excluded. Activity abundance as used here is a combination of actual abundance in the habitat and a measure of activity since more active species are likely to be captured more often and so appear more abundant. Pitfall traps are designed to capture solitary, wandering arthropods [25]. Spiders are often rare, analysis at the species level results in matrices with many zeroes, therefore, I lumped species by genus for analysis. In cases where there was only one species for a given genus, the analysis is at the species level. This also allowed me to include juvenile stages that could be identified to genus, even if not to species. I excluded juveniles that could be identified only to family. If only one species was present, the juvenile stages were included in the count for the adults (e.g., the gnaphosid *Drassodes neglectus*). Whether or not to include juvenile stages is a long-standing issue with arachnids [26]. While they are not the same as adults, they are an important part of the spider biomass present at almost all collecting times, and they are the stages that readily disperse [9] and would recolonize burned areas. The two harvestman species (*Togwoteeus biceps* and *Leiobunum* sp.) and their immatures were included in the analyses. Mites that were included were divided into the following higher-taxonomic groups: Mesostigmata; Oribatida (the moss mites or beetle mites); and Trombidiformes, including Anystidae, Bdellidae, and a few undetermined groups. The Erythraeidae were most common. They are predators in adult and nymphal stages but as larvae are parasitic on other arthropods [27].

In PRIMER [28], species accumulation curves for the observed number of species, using Chao2 and Jackknife 2 estimators, showed sampling effort that was underestimating richness initially but approximated the estimated richness over the time course of the study (S1). Pairwise comparisons of sites within treatment and habitat were not significantly different in R's adonis program [29].

Because sample collection dates varied, I grouped collection periods by season: spring (May–June), monsoon (July–September), and fall (October–November). Traps were open continuously from spring through fall; they were closed and often snow-covered in winter. Analyses were run at the site level.

I did not compare across habitats because the trap capture probability differed by habitat based on differences in amounts of open ground and litter [30] and the fire severity differed. I followed the same analyses for each habitat. Because the abundance levels varied widely, abundance numbers were log-transformed in PRIMER and R and grouped at the site level. In PRIMER, I used PERMANOVA to test for treatment (burned or unburned), year, and treatment x year effects. I used adonis in R for pairwise treatment and year comparisons of activity abundance for each habitat. I tested for autocorrelation (autocorr in R) over the years. Both PERMANOVA and autocorr took into account that the samples were collected from the same traps over time. I used non-metric multidimensional scaling (NMDS in R and in PRIMER, Bray-Curtis distances) to visualize the PERMANOVA results for treatment and year.

In PRIMER I used Similarity Percentages (SIMPER), which compared Bray–Curtis dissimilarities in the mean between-group differences in activity abundance among samples by treatment and among years. The method can be influenced by a large variation within the group. I also used this to show the percent contribution of the most abundant genera for treatment and year. Indicator genera (indicspp in R) showed taxa that were associated with a particular treatment or year.

## 3. Results

Spider taxa used in analyses of the monitoring effort were: 64 species in 29 genera and 10 families for PP; 69 species in 34 genera in 10 families for MC; and 58 species in 37 genera in 10 families for MV. The common co-dominant spider families in temperate areas, Lycosidae and Gnaphosidae, along with harvestmen and the erythraeid mites, made up about 60% of the abundance (Tables A1–A3).

Even with uneven sampling, PERMANOVA effects of wildfire and year were significant in all three habitats, even for the lightly burned MV grassland. The interaction between fire and year was significant for PP and MC, but not for MV (Table 1). These effects were visualized with NMDS

by treatment and year (Table 2, Figures 4–6). Besides losing tree canopy cover, on the ground, the forest habitats lost litter cover, much of which was replaced with cover by grasses and herbaceous plants (Figure 2). The cover that had previously been there included pine needles, dead aspen leaves, and downed tree branches; a quite different microhabitat for small ground-dwelling arachnids. The directions of change within the habitats were similar in that the NMDS ellipses from 2012–2015 overlap (Figures 4–6), but autocorrelation analysis showed that year was important only as a factor and not as a numerical time sequence. Arachnid abundance varied over three orders of magnitude, although most samples contained 10–40 individuals. The wolf spider *Pardosa* was the dominant genus in all habitats, frequently the highest contributor to abundance and never falling below the top 5 ranked taxa (Table 3). *Pardosa* remained the dominant genus; 5 of the 7 species were found in all 3 habitats (Tables A1–A3). At PP, the wolf spider *P. distincta* was most abundant; all species except *P. yavapa* were collected mostly from burned sites. At MC, *P. uncata*, common in conifer forests [31], was most abundant in both burned and unburned sites. At MV, *P. distincta* and *P. concinna* were most abundant in burned and unburned sites; *P. yavapa* and *P. uncata* were the two species not found there. The species generally overlapped in time: *P. coloradensis* and *P. concinna* were adults in the spring and continued into the summer, while the other species were more abundant in the summer. *Pardosa xerophila* and *P. montgomeryi* were collected in rather low numbers in the forest habitats, mostly in burned sites. At MV, *P. montgomeryi* was collected more often than *P. xerophila*. Few *Pardosa* adults were present in fall collections.

**Table 1.** PERMANOVA results for arachnid genera (and some higher taxa) for effects of fire, year, and their interaction for ponderosa pine forest, mixed-conifer forest, and mountain valley grassland at the Valles Caldera. ** 0.01> $p$ >0.001, *** $p$ = 0.001, n.s. = not significant.

| Source | df | Pseudo-F | $p$-Value |
|---|---|---|---|
| Ponderosa Pine Forest (PP) | | | |
| Fire | 1 | 12.061 | 0.007 ** |
| Year | 4 | 3.4166 | 0.001 *** |
| Fire × Year | 4 | 2.2966 | 0.001 *** |
| Mixed-conifer Forest (MC) | | | |
| Fire | 1 | 7.3596 | 0.009 ** |
| Year | 4 | 7.1894 | 0.001 *** |
| Fire × Year | 4 | 3.164 | 0.001 *** |
| Mountain Valley Grassland (MV) | | | |
| Fire | 1 | 12.128 | 0.009 ** |
| Year | 4 | 3.7021 | 0.001 *** |
| Fire × Year | 4 | 0.5723 | 0.985 n.s. |

**Table 2.** Non-metric multidimensional scaling (NMDS) results for ponderosa pine forest (PP), mixed-conifer forest (MC), and mountain valley grassland (MV) at the Valles Caldera National Preserve by treatment (burned/unburned) and years. Significance values: * 0.05 > $p$ >0.01, ** 0.01 > $p$ > 0.001, *** $p$ = 0.001, n.s. not significant.

| Habitat | PP | MC | MV |
|---|---|---|---|
| No. of runs | 500 | 500 | 800 |
| 2-D Stress | 0.26 | 0.28 | 0.28 |
| Goodness of fit Trmt $R^2$ | 0.3253 ** | 0.2423 *** | 0.0545 ** |
| Goodness of fit Year $R^2$ | 0.1261 ** | 0.1049 *** | 0.0331 n.s. |
| Non-metric fit $R^2$ | 0.935 | 0.921 | 0.925 |
| Linear fit $R^2$ | 0.668 | 0.610 | 0.658 |
| 3-D Stress | 0.18 | 0.20 | 0.20 |
| Goodness of fit Trmt $R^2$ | 0.3256 ** | 0.2574 ** | 0.0692 ** |
| Goodness of fit Year $R^2$ | 0.1227 ** | 0.0975 ** | 0.0531 * |
| Non-metric fit $R^2$ | 0.968 | 0.961 | 0.958 |
| Linear fit $R^2$ | 0.760 | 0.714 | 0.728 |

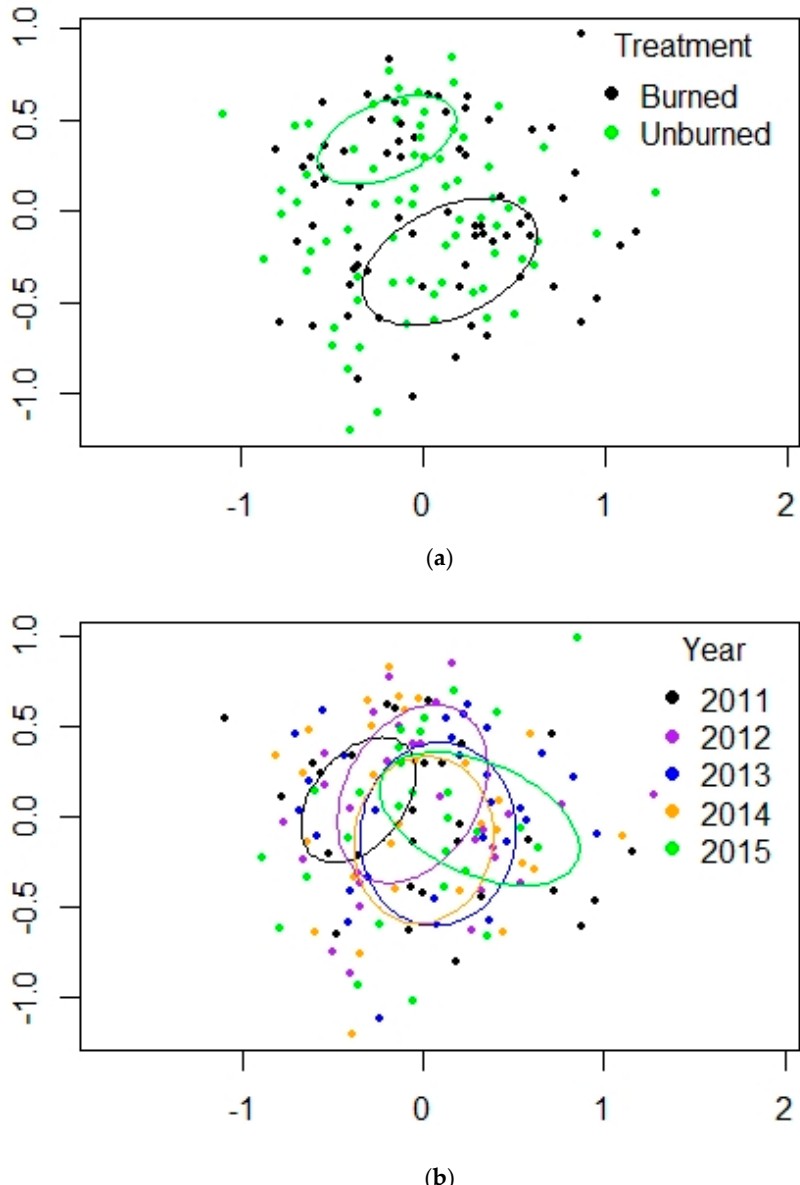

**Figure 4.** (**a**,**b**). NMDS plots for PERMANOVA results for ponderosa pine forest at Valles Caldera. (**a**) treatment, (**b**) year. Ellipses are the 95% confidence intervals for the locations of the centroids.

### 3.1. Ponderosa Pine Forest (PP)

In this habitat, the fire was severe enough to be stand-replacing; this area may become and remain a shrubfield for many years [20,32]. SIMPER results for the arachnids showed average similarity ranged from 28.77 to 39.33 across the five years, and the unburned sites were more similar to each other (42.14) than the burned ones (34.02). The top 5 taxa at PP accounted for 66.47% of abundance at burned sites and 70.78% at unburned sites and ranged from 63.17% to 69.68% across the 5 years of the study in percent contribution to activity abundance. Besides the high numbers of lycosids and gnaphosids, other families were also important: Dictynidae (*Cicurina*), Linyphiidae (*Erigone*, *Agyneta*), Sclerosomatidae (*Togwoteeus*), and Erythraeidae. Abundance was lowered from 2013–2015 because of the loss of 3 unburned sites to the Thompson Ridge Fire, as noted in the Sites and Methods section above. The number of individuals in samples varied greatly, but there was a general trend of increasing abundance in burned and unburned plots from 2011–2015. For the PP habitat, pairwise adonis tests of differences among pairs of years showed significant values for 2011 vs. 2012 ($p = 0.0108$), 2011 vs. 2013 ($p = 0.0004$), and 2013 vs. 2014 ($p = 0.0133$). All other paired comparisons for the year were not

significant. Even though treatment differences were significant, there was overlap between the two over the 5 years (Figure S2a). PERMANOVA results and the NMDS visualization showed that the effect of the fire on arachnids was strong; the ellipses for the 95% confidence intervals for burned and unburned sites were widely separated, and the years remained relatively distinct, with increasing distance between the community of 2011 and 2015 (Table 1, Table 2, Figure 4a,b). Because of the amount of scatter in the 2-D plots, I also included the stress and fit values for the 3-D NMDS, which were an improvement in capturing the variability of the samples. However, the $R^2$ values for the goodness of fit for treatment and year were not much changed.

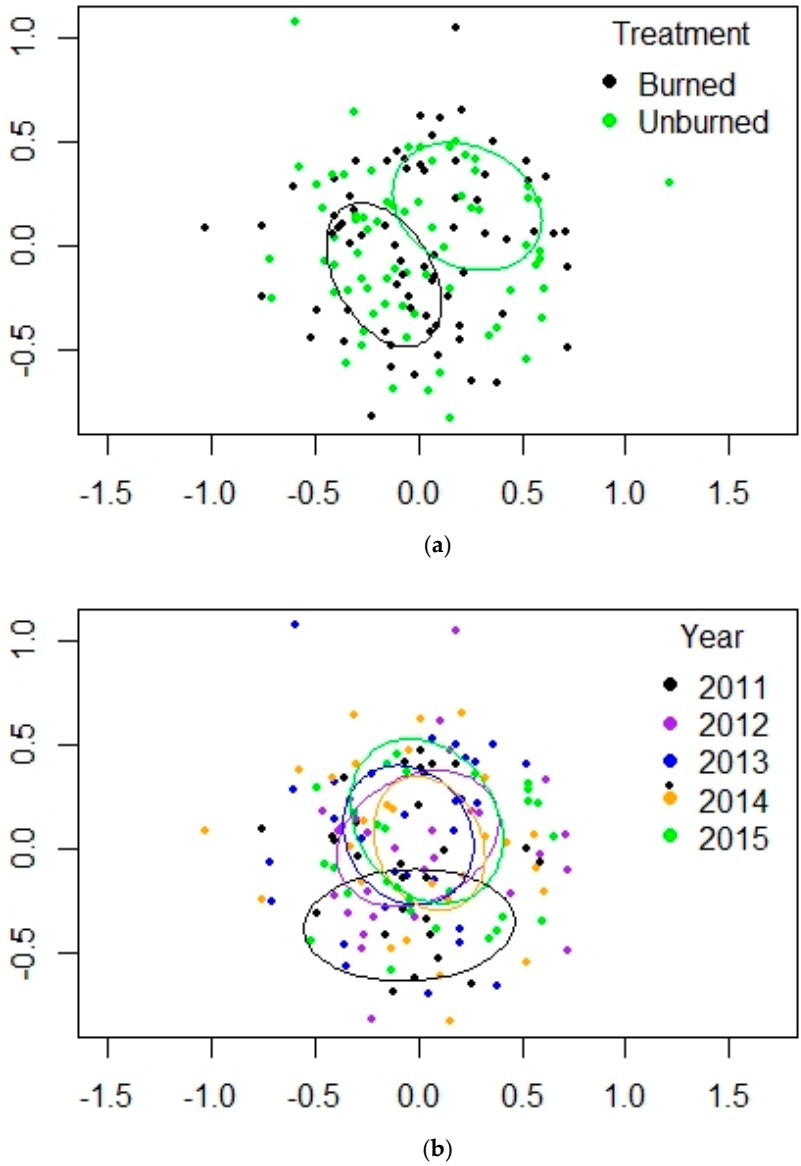

**Figure 5.** (**a**,**b**). NMDS plots for PERMANOVA results for mixed-conifer forest at Valles Caldera. (**a**) treatment, (**b**) year. Ellipses are the 95% confidence intervals for the locations of the centroids.

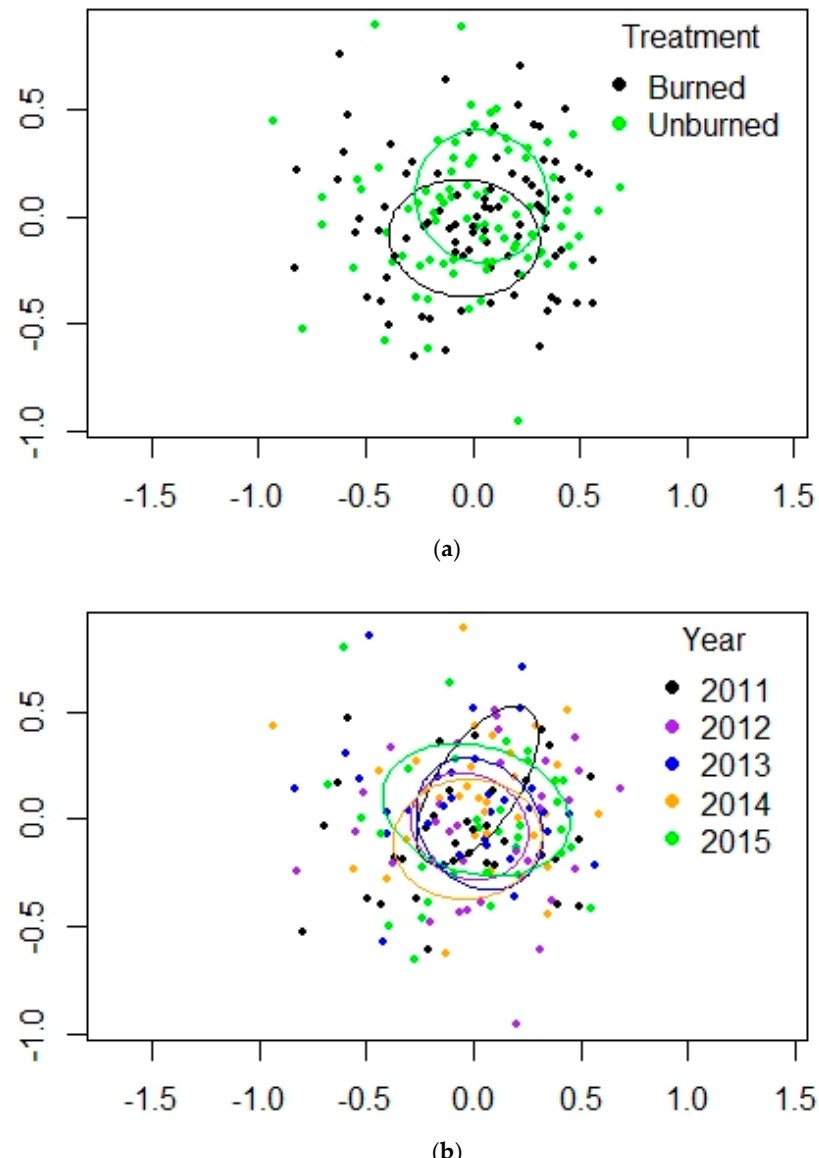

**Figure 6.** (**a**,**b**). NMDS plots for PERMANOVA results for mountain valley grassland at Valles Caldera. (**a**) treatment, (**b**) year. Ellipses are the 95% confidence intervals for the locations of the centroids.

In contrast, indicator analysis examined genera that contributed to the difference between burned and unburned areas. Few genera characterized the burned sites (Table 4). The linyphiid spider *Erigone dentosa* showed an unusual increase in the number of individuals in the burned sites from 0 in 2011, to 8 in 2012, to 148 in 2013, to 513 in 2014, and 253 in 2015 (Table A1 in Appendix A). Linyphiid spiders were speciose but often with few individuals in samples, as is common for this family [12]. Erythraeidae were also significant indicators of burned sites. The unburned sites had many more indicator genera across six spider families, one harvestman family, and several mite taxa. The dominant genera *Pardosa* wolf spider and *Haplodrassus* and *Gnaphosa* (both ground spiders) were generally not displaced (Table 3), but the unburned sites were clearly richer in genera, especially among the Linyphiidae, and among other lycosid and gnaphosid genera. The harvestman (*Leiobunum*), predatory mites (especially the Anystidae), and the beetle mites (Oribatida) were also important in distinguishing the unburned sites (Table 4).

**Table 3.** Arachnid community SIMPER results by treatment and year for Ponderosa Pine Forest, Mixed-conifer Forest, and Mountain Valley Grassland at the Valles Caldera National Preserve.

| | Ponderosa Pine Forest | | Mixed-Conifer Forest | | Mountain Valley Grassland | |
|---|---|---|---|---|---|---|
| Factor | Average Similarity | Top 5 Taxa % Contribution | Average Similarity | Top 5 Taxa % Contribution | Average Similarity | Top 5 Taxa % Contribution |
| Burned | 34.02 | *Pardosa* 22.95 | 39.47 | *Pardosa* 20.95 | 52.11 | *Pardosa* 27.56 |
| | | *Haplodrassus* 13.96 | | *Gnaphosa* 16.80 | | Erythraeidae 25.20 |
| | | Erythraeidae 10.39 | | *Cicurina* 13.76 | | *Haplodrassus* 13.80 |
| | | *Cicurina* 9.89 | | *Haplodrassus* 12.26 | | *Xysticus* 11.23 |
| | | *Erigone* 9.28 | | *Togwoteeus* 9.52 | | *Thanatus* 6.50 |
| | | Cumulative 66.47% | | Cumulative 73.30% | | Cumulative 84.30% |
| Unburned | 42.14 | *Anyphaena* 16.24 | 40.44 | *Leptobunus* 18.32 | 51.95 | *Pardosa* 30.39 |
| | | *Pardosa* 16.23 | | *Togowteeus* 17.85 | | *Haplodrassus* 13.75 |
| | | *Gnaphosa* 14.00 | | *Pardosa* 12.85 | | Erythraeidae 13.22 |
| | | *Alopecosa* 13.17 | | *Haplodrassus* 9.03 | | *Xysticus* 12.93 |
| | | *Haplodrassus* 11.14 | | *Cicurina* 6.51 | | *Grammonota* 10.35 |
| | | Cumulative 70.78% | | Cumulative 64.55% | | Cumulative 80.64% |
| Year 2011 | 39.33 | *Haplodrassus* 28.14 | 37.84 | *Pardosa* 26.55 | 55.08 | *Pardosa* 23.88 |
| | | *Gnaphosa* 15.51 | | *Haplodrassus* 24.07 | | *Haplodrassus* 18.78 |
| | | *Pardosa* 11.03 | | *Cicurina* 18.11 | | Erythraeidae 18.26 |
| | | Erythraeidae 7.22 | | *Gnaphosa* 8.32 | | *Xysticus* 12.96 |
| | | *Alopecosa* 6.47 | | *Mesmessus* 5.22 | | *Thanatus* 7.51 |
| | | Cumulaltive 68.37% | | Cumulative 81.97 | | Cumulative 81.40% |
| Year 2012 | 32.31 | *Haplodrassus* 23.88 | 39.57 | *Togwoteeus* 17.54 | 57.77 | *Pardosa* 30.26 |
| | | *Pardosa* 18.09 | | *Leiobunum* 15.03 | | *Haplodrassus* 17.80 |
| | | *Gnaphosa* 13.87 | | *Gnaphosa* 14.92 | | Erythraeidae 14.90 |
| | | *Alopecosa* 8.73 | | *Haplodrassus* 11.17 | | *Xysticus* 13.45 |
| | | Erythraeidae 5.12 | | *Pardosa* 9.72 | | *Thanatus* 7.18 |
| | | Cumulative 69.68% | | Cumulative 68.37% | | Cumulative 83.59% |
| Year 2013 | 28.77 | *Pardosa* 20.45 | 39.82 | *Pardosa* 14.77 | 52.60 | *Pardosa* 26.03 |
| | | *Gnaphosa* 13.08 | | *Gnaphosa* 14.75 | | Erythraeidae 21.62 |
| | | *Haplodrasssus* 11.44 | | *Togwoteeus* 14.36 | | *Haplodrassus* 13.49 |
| | | *Erigone* 10.34 | | *Leiobunum* 10.18 | | *Xysticus* 12.12 |
| | | Erythraeidae 7.85 | | *Cicurina* 9.62 | | *Thanatus* 8.38 |
| | | Cumulative 63.17% | | Cumulative 63.69% | | Cumulative 81.64% |
| Year 2014 | 38.39 | *Pardosa* 24.84 | 43.68 | *Pardosa* 18.25 | 51.02 | *Pardosa* 31.28 |
| | | *Cicurina* 14.66 | | *Togwoteeus* 14.54 | | Erythraeidae 17.67 |
| | | *Erigone* 10.20 | | *Cicurina* 12.00 | | *Haplodrassus* 12.69 |
| | | *Gnaphosa* 8.03 | | *Haplodrassus* 11.01 | | *Xysticus* 12.35 |
| | | *Togwoteeus* 6.98 | | *Gnaphosa* 9.67 | | *Thanatus* 5.33 |
| | | Cumulative 64.72% | | Cumulative 65.47% | | Cumulative 79.32% |
| Year 2015 | 34.36 | *Pardosa* 40.49 | 40.07 | *Pardosa* 27.13 | 48.49 | *Pardosa* 30.18 |
| | | *Haplodrassus* 6.23 | | *Togwoteeus* 20.77 | | Erythraeidae 19.29 |
| | | *Agyneta* 6.08 | | *Gnaphosa* 9.84 | | *Grammonota* 9.89 |
| | | Erythraeidae 6.05 | | *Erigone* 8.62 | | *Xysticus* 8.83 |
| | | *Togwoteeus* 5.98 | | *Leiobunum* 8.54 | | *Haplodrassus* 8.22 |
| | | Cumulative 64.84% | | Cumulative 74.90% | | Cumulative 76.42% |

**Table 4.** Indicator taxon analysis of arachnids for treatment and year for Ponderosa Pine Forest, Mixed-conifer Forest, and Mountain Valley Grassland at Valles Caldera National Preserve. Significance values: * $0.05 > p > 0.01$, ** $0.01 > p > 0.001$, *** $p = 0.001$. If a year is not listed, there was no indicator taxon for it.

| Habitat | Factor | Family (or Higher Taxon) | Genus | Indicator Value |
|---|---|---|---|---|
| Ponderosa Pine Forest | Burned | Gnaphosidae | *Micaria* | 0.300 *** |
| | | Linyphiidae | *Erigone* | 0.458 *** |
| | | | *Grammonota* | 0.345 *** |
| | | | *Islandiana* | 0.319 *** |
| | | Erythraeidae | | 0.278 ** |
| | Unburned | Anyphaenidae | *Anyphaena* | 0.586 *** |
| | | Dictynidae | *Hackmania* | 0.179 * |
| | | Gnaphosidae | *Drassodes* | 0.179 * |
| | | | *Gnaphosa* | 0.425 *** |
| | | | *Zelotes* | 0.219 ** |
| | | Linyphiidae | *Helophora* | 0.239 *** |
| | | | *Incestophantes* | 0.261 *** |
| | | | *Lepthyphantes* | 0.177 * |
| | | | *Pocadicnemis* | 0.350 *** |
| | | | *Spirembolus* | 0.328 *** |
| | | | *Tachygyna* | 0.317 *** |
| | | | *Walckenaeria* | 0.293 *** |
| | | | *Wubana* | 0.448 *** |

**Table 4.** *Cont.*

| Habitat | Factor | Family (or Higher Taxon) | Genus | Indicator Value |
|---|---|---|---|---|
| | | Lycosidae | *Alopecosa* | 0.549 *** |
| | | | *Hogna* | 0.176 * |
| | | Thomisidae | *Xysticus* | 0.246 ** |
| | | Sclerosomatidae | *Leiobunum* | 0.169 * |
| | | Oribatida | | 0.294 *** |
| | | Trombidiformes | Undetermined | 0.175 * |
| | | Anystidae | | 0.293 *** |
| | Year 2011 | Clubionidae | *Clubiona* | 0.294 ** |
| | | Gnaphosidae | *Haplodrassus* | 0.394 *** |
| | | Hahniidae | *Neoantistea* | 0.298 *** |
| | | Phrurolithidae | *Scotinella* | 0.282 ** |
| | Year 2014 | Dictynidae | *Cicurina* | 0.352 ** |
| | Year 2015 | Linyphiidae | *Grammonota* | 0.362 *** |
| | | Mesostigmata | | 0.402 *** |
| Mixed-conifer Forest | Burned | Linyphiidae | *Erigone* | 0.348 *** |
| | | | *Mermessus* | 0.348 *** |
| | | Phrurolithidae | *Phrurolithus* | 0.230 ** |
| | Unburned | Amaurobiidae | *Callobius* | 0.282 *** |
| | | Gnaphosidae | *Orodrassus* | 0.228 ** |
| | | Linyphiidae | *Ceratinella* | 0.268 *** |
| | | | *Helophora* | 0.397 *** |
| | | | *Islandiana* | 0.244 *** |
| | | | *Pocadicnemis* | 0.201 ** |
| | | | *Sisicottus* | 0.350 *** |
| | | | *Spirembolus* | 0.216 ** |
| | | | *Tachygyna* | 0.358 *** |
| | | | *Tapinocyba* | 0.196 * |
| | | | *Wubana* | 0.466 *** |
| | | Lycosidae | *Trochosa* | 0.252 *** |
| | | Salticidae | *Pelegrina* | 0.198 ** |
| | | Theridiidae | *Steatoda* | 0.289 *** |
| | | Sclerosomatidae | *Leiobunum* | 0.543 *** |
| | | | *Togwoteeus* | 0.359 *** |
| | | Oribatida | | 0.413 *** |
| | | Trombidiformes | Undetermined | 0.183 ** |
| | | Anystidae | | 0.288 *** |
| | | Bdellidae | | 0.317 *** |
| | Year 2013 | Salticidae | *Pelegrina* | 0.272 *** |
| | Year 2015 | Linyphiidae | *Grammonota* | 0.440 *** |
| | | Lycosidae | *Pardosa* | 0.428 *** |
| | | Mesostigmata | | 0.353 *** |
| Mountain Valley Grassland | Burned | Gnaphosidae | *Micaria* | 0.195 ** |
| | | Linyphiidae | *Tachygyna* | 0.175 * |
| | | Erythraeidae | | 0.282 ** |
| | Unburned | Dictynidae | *Cicurina* | 0.224 ** |
| | | Gnaphosidae | *Gnaphosa* | 0.238 *** |
| | | Hahniidae | *Neoantistea* | 0.187 * |
| | | Linyphiidae | *Grammonota* | 0.390 *** |
| | | | *Walckenaeria* | 0.298 *** |
| | | Philodromidae | *Ebo* | 0.206 ** |
| | | Sclerosomatidae | *Leiobunum* | 0.148 ** |
| | Year 2011 | Gnaphosidae | *Haplodrassus* | 0.316 *** |
| | | Salticidae | *Pellenes* | 0.286 ** |
| | | Anystidae | | 0.275 ** |
| | Year 2013 | Linyphiidae | *Tachygyna* | 0.331 ** |
| | Year 2014 | Linyphiidae | *Tapinocyba* | 0.247 * |
| | | | *Walckenaeria* | 0.222 * |
| | Year 2015 | Linyphiidae | *Erigone* | 0.460 *** |

### 3.2. Mixed-Conifer Forest (MC)

Fire severity ranged from moderate to severe in this forest. SIMPER showed arachnid similarity values of 37.84–40.07 across the five study years, and within treatments the similarities were close: 39.47 for burned and 40.44 for unburned (Table 3). The top 5 taxa at MC accounted for 73.70% of abundance at burned sites and 64.55% at unburned sites, and their contribution to activity abundance ranged from

63.69% to 81.97% across the 5 years of the study, a wider range than at PP. Besides the expected high numbers of lycosids and gnaphosids, 2 other spider families were also important: Dictynidae (*Cicurina*) and Linyphiidae (*Erigone*). The 2 harvestmen species *T. biceps* and *Leiobunum* sp. were dominants in unburned areas; the high abundance levels for 2012–2013 in unburned areas are largely due to these two species (Table A2 in Appendix A). The linyphiid *E. dentosa* also appeared in large numbers in burned sites (403 in 2014 and 324 in 2015), although lower than at PP (Tables A1 and A2). Arachnid abundance was lower at burned sites through 2013 but increased steadily through 2015. For the MC habitat, pairwise adonis tests of differences among pairs of years showed significant values only for 2012 vs. 2015 (*p* = 0.0477). All other paired comparisons for the year were not significant. The scatter among the data points was less than at PP, which made comparisons by treatment among years less clear (Figure S2b). PERMANOVA results were significant for treatment, year, and the interaction of the two factors (Table 1). The NMDS plots illustrated clear separation between treatments, and 2011 was quite different from the remaining years (Table 2, Figure 5a,b); the 3-D NMDS improved the stress but did not much change the $R^2$ for treatment or year. There were only three indicator genera for the burned sites at MC, including *E. dentosa* (Table 4). For the unburned sites, there were 20 indicator genera, including 9 linyphiids, both harvestmen genera, and four mite groups. The dominant genera *Pardosa*, *Haplodrassus*, and *Gnaphosa* were consistently in the top 5 taxa (Table 3). The unburned sites contained more genera, especially among the Linyphiidae. The harvestman (*Leiobunum* sp.), predatory mites (especially the Anystidae), and the beetle mites (Oribatida) were also important in distinguishing the unburned sites (Table 4).

### 3.3. Mountain Valley Grassland (MV)

The burn in this grassland habitat was of low severity. Within-group similarities for treatment and across the years were the highest of the three habitats, from 48.49 to 57.77 (Table 3). *Pardosa* ranked first for each year and both treatments, contributing 23–31% of the abundance. The gnaphosid *Haplodrassus* was also in the top 5 taxa for treatment and year, contributing 8–19% of abundance. The linyphiid *E. dentosa* appeared in large numbers only in 2015, in both burned and unburned sites (Table A3 in Appendix A). The harvestman *T. biceps* occurred in very low numbers here. Erythraeid mites were in the top 5 taxa for treatment and year, contributing 13–22% of abundance. Two genera were common at MV that were rare or not collected at PP or MC: the crab spider (Thomisidae) genus *Xysticus* (in low numbers in the forest sites) and the philodromid genus *Thanatus* (not collected at MC) (Tables A1–A3). In every year, some samples contained well over 100 arachnids; samples at PP and MC only approached those numbers in 2014–2015, whether in burned or unburned sites. Recall that at PP and MC, 3 of the unburned sites were not included from 2013–2015 because they burned in the Thompson Ridge fire. For the MV habitat, pairwise adonis tests of differences among pairs of years showed significant values only for 2012 v. 2015 (*p* = 0.0477). All other paired comparisons for years were not significant. The NMDS showed greater overlap between the burned and unburned treatments (Figure 6) and the scatterplot of treatment and years showed several points from 2014 and 2015 farther from the rest (Figure S2).

PERMANOVA results were significant for treatment and year but not the interaction factor (Table 1). NMDS results showed the greatest amount of overlap of the three habitats, but 2011 was still somewhat different from the other years (Table 2, Figure 6a,b). There was less difference between burned and unburned sites, but the 3-D NMDS was able to reduce the stress and showed a significant effect of year. There were few indicator genera for treatment at MV. Seven taxa were indicators for the unburned sites, including the harvestman *Leiobunum* sp. and the philodromid genus *Ebo*, which was not collected at PP or MC.

## 4. Discussion

The arachnids within three habitats at the Caldera that were affected by the Las Conchas wildfire were similar in their responses (the ellipses largely overlapped) (Figures 4–6), perhaps because many

of the genera and species were shared. However, vegetation structure in each habitat was different, so that pitfall trap capture probabilities were not comparable across the three. Abundance returned to pre-fire levels in about 3 years (similar to results in Abbott et al. [33] in the Jarrah forest in Australia, Smith DiCarlo et al. [18] in Oregon grasslands, and Samu et al. in the Hungarian juniper-poplar steppe [17]). Nevertheless, at the level of species composition, there were a number of changes, even in unburned areas. Many spider species are rare in pitfall collections [16,30] (Tables A1–A3), and so are not sampled each year, which can complicate what recovery means and makes long-term monitoring important for getting a more complete picture of species occurrence in an area. For example, over the five years of this study, the abundance of *Pardosa*, the dominant genus among ground-dwelling spiders, varied from 11% to 40% at PP, 5–24% at MC, but was more consistent in the lightly burned MV at 24–30% (Table 3). Interactions with other taxa are also likely to be important. The gnaphosid genus *Micaria* is well known for ant mimicry, both in associating with ants and preying on them [34]. Knudsen [35] reported increased ant abundance in PP burned sites after Spring 2012, which may have influenced *Micaria* activity abundance as well. For other less-abundant taxa, longer time scales may be needed to discern patterns in abundance and richness, particularly with influences from disturbance, such as fire or drought (for forests [6,33,36]; for grasslands [37]). It may turn out that there is no long-term pattern because the disturbances occur in a context of weather and vegetation structure that are themselves changing [4,32]. From 2011–2015, spring and fall precipitation at the Caldera was variable; every year showed different patterns and the summer monsoon period was delayed in 2011 and 2013. As a result, year was important as a factor but not as a time series influencing the arachnid community (Figures 4–6), as was also the case in Oregon forest [12] and Australian grasslands [33]. At the Caldera there was considerable variation in numbers of individuals in samples across the years; often both burned and unburned sites increased or decreased together, suggesting annual weather was as big a factor as vegetation changes resulting from the fire.

Taxon dominance patterns at the Caldera were not greatly disrupted (Table 3). What affected more were the small litter-dwellers (such as *Helophora*, *Sisicottus*, and *Wubana* in the Linyphiidae, and *Neoantistea* in the Hahniidae), which made up a large part of the indicators for unburned sites (Table 4). Their numbers were generally low so that 5 years post-fire were not enough to say if they were moving back into the burned sites (Tables A1–A3). Gillette et al. [13] found greater declines in spider abundance and richness in California forest stands that were structurally diverse, suggesting that the higher fuel loads there might require longer recovery times for the vegetation structure and the spider community. Lycosids and gnaphosids were common in burned areas in forests in Oregon [12] and Alberta, Canada [36] up to 15 years after a fire. Linyphiidae and Thomisidae were more common in unburned areas there, which matches the patterns found in the Caldera even after 5 years. It is unclear whether herbaceous and grass vegetation is acceptable for the species that had previously lived under the tree canopy and in the litter. Some of the philodromids, such as members of the genus *Ebo* [38], are common in grassy areas [10], yet did not increase in numbers in the forest sites, even though forb and grass cover increased there. Over the course of the study, there was no trend that the spider communities in the burned forest sites were becoming more like the grassland sites. In German forests of spruce and Douglas-fir, Ziesche and Roth [39] found tree canopy and litter were important to ground-dwelling spiders overall, but that microhabitats varied within forest stands, allowing the spiders to move around over the course of a year, as the seasons changed. The linyphiid spiders are a large family of small-bodied spiders that are generally rare at the species level and strongly associated with litter and low vegetation [10]. In Oregon forest [12], the linyphiids made up most of the species richness, but there was no dominant species because abundances were low. Linyphiids were dominants in Finnish forests 3 years post-fire [15], and Buddle et al. [36] showed linyphiids in Canadian forest were abundant in areas that had not burned for decades. At the Caldera, herbaceous cover increased and litter cover correspondingly decreased (Figure 2) at the burned sites, but the linyphiids that were indicator species of unburned areas did not move to burned areas over the course of the study, suggesting that litter and tree canopy cover provided important microhabitat

in ways that herbaceous plants did not. Because of their small size (generally < 5 mm), they can disperse readily by ballooning [9], so that it seemed unlikely that the spiders were unable to colonize the burned areas. Hundreds of *E. dentosa* were collected in burned areas in PP (in 2013–2015), MC, and MV (in 2014–2015) (Tables A1–A3). At PP and MC, numbers were lower in 2015, suggesting that the high numbers of *E. dentosa* were a short-lived occurrence, another reason for multi-year studies that can capture such transient changes. Members of this family are understudied in the southwestern US; therefore, the specimens and records from the Caldera are important additions to the knowledge of regional distribution and habitat association. While most of the linyphiids were indicators of unburned conditions, there were a few besides *E. dentosa* that were more abundant in burned areas, such as *Islandiana flaveola* at PP and *Mermessus taibo* at MC (Tables A1 and A2).

Spiders made up the majority of the arachnid taxa, but harvestmen and mites also figured prominently in the top 5 of abundant taxa at PP and MC. New Mexico is in the southern part of the range of the harvestman *T. biceps*, where it occurs at high elevations, from dense forest to above treeline [40]. *Leiobunum* sp. occurred in comparable numbers to *T. biceps* at MC but was less abundant at PP (Tables A1 and A2). Both species were rare at MV, but *Leiobunum* sp. was an indicator of unburned sites there. Pitfall traps do not sample mites fully but at least collect those active on the ground surface (see Reference [41] for results from Berlese funnel extractions). The erythraeids were numerous (Tables A1–A3) and were indicators in burned sites at all three habitats. In ponderosa pine forest in California [41], one year following the prescribed fire, mites from litter and loose soil declined across all groups, except Prostigmata (now in Trombidiformes). Mites from pitfall samples in the same area collected in the following year had increased in abundance [13]. In Swedish boreal forest, Malmstrom et al. [14] found that the Trombidiformes needed only a year or so to return to pre-fire abundance levels, but that the Mesostigmata needed about 5 years. At the Caldera, results were similar, with Mesostigmata numbers low until 2015 in burned areas in all three habitats (Tables A1–A3). Possible reasons for the longer recovery time include life history features, such as low reproductive or growth rates. For some soil-dwelling mesostigmatans, Krantz and David [27] reported that the females spend considerable time and energy finding plant rootlets to lay their eggs on, so the larvae hatch in a place where their symphylan prey occur.

Anystid (whirligig mites) and bdellid (snout mites) numbers were generally low, but they were indicators for unburned sites (Table 4). These taxa are rarely reported, but make up a part of the predator trophic level of surface-active arthropods.

There was a high overlap at the family and genus level among the habitats; most families occurred in each (but sometimes not in high enough numbers to be included in these analyses), except that Amaurobiidae were only at MC and Anyphaenidae were only at PP and MC. Amaurobiids occur in dense forests with tree canopy cover and downed trees [10]. *Anyphaena marginalis* is common in pine/oak forests of the southwestern US [42] and was collected mostly at PP, with few in MC. Samu et al. [17] and Smith DiCarlo et al. [18] found high overlap at their sites so that analysis at higher taxonomic levels masked differences at the species level. Within the spiders, many species are prey generalists but are strongly specialized in habitat needs, whether for web construction (the styles are extremely variable), or for temperature or humidity preferences, or for sites for depositing egg sacs. Nevertheless, suitable microhabitats are often available to them within broader habitat types [39], but these may come and go with disturbance, such as fire or drought. Studies assessing the extent of determinism in structuring spider assemblages have had mixed results. In grassland, Langlands et al. [37] suggested that species composition in burned areas over time did come to resemble the unburned species composition, but it took at least 10 years in that arid system to begin to see that pattern, and different sites could take different paths to reach the unburned condition. Ferrenberg et al. [43] worked with post-fire arthropod assemblages in the Jemez Mountains, including the Caldera. Their question was also about determinism in arthropod assemblages in the context of the number of fires and time since fire. Those factors were significant, but the habitat features that were measured did not strongly correspond to the indicator taxa. It is possible that they were not "seeing the habitat from the spiders' point of

view," to know which habitat features were important or at what scale. Like Langlands et al. [37], they suggested that both stochastic and deterministic processes occur in the communities but are of variable importance. In the short term, the changes in abundance appeared to be stochastic, particularly in the context of variable weather conditions from one year to the next.

With fire returning more frequently and more severely to forests of the southwestern U.S. [2,4], arachnid communities are "reset" over and over, with little chance to develop their long-term patterns, which may require up to 15+ years [6,36–38]. Some areas of ponderosa pine forest that have undergone stand-replacing fire may persist as shrubfields of oak or other species for many decades [20,31]. For the arachnids of the Caldera, this could mean loss of rarer species locally for a time, even though a higher taxon analysis might show that richness and abundance were generally not changing. Nevertheless, for centuries, fire has been one of the producers of the Caldera landscape mosaic and the inhabitants are accustomed to the variation in past climate factors. The main concern is that future changes may well be outside the historic range of conditions [19], having an outsized impact on species that are poorly known to begin with, such as arthropods that are small, cryptic, or strongly seasonal [44].

**Supplementary Materials:** The following are available online at http://www.mdpi.com/1424-2818/12/10/396/s1, Figure S1: Species accumulation curves for samples in (**a**) ponderosa pine forest (PP), (**b**) mixed-conifer forest (MC), and (**c**) mountain valley grassland (MV) at the Valles Caldera National Preserve. Sobs is the observed number of taxa compared with the estimators Chao2 and Jackknife 2. Figure S2: NMDS scatterplots from PRIMER by treatment (**B** = Burned, **U** = Unburned) and year for (**a**) ponderosa pine forest (PP), (**b**) mixed-conifer forest (MC), and (**c**) mountain valley grassland (MV) at the Valles Caldera National Preserve, New Mexico, USA. Figure S3: R code for NMDS, autocorr, and indicspp.

**Funding:** This research was funded by the Collaborative Forest Landscape Restoration Project (CFLRP).

**Acknowledgments:** My sincere thanks go to Robert Parmenter, chief scientist at the Valles Caldera National Preserve for including post-fire arthropod monitoring and helpful comments on the manuscript; Mark Ward, field work and manager of sample sorting, and field crews; Martina Suazo for vegetation data; field and lab crews, especially Joaquin Garcia, Lozen Benson, and Marlo McCarter; and to J. Rudgers for her R class. I am grateful to the reviewers for their thorough reading and thoughtful comments on the manuscript. This paper is in memory of Dr. Norman I. Platnick (1951–2020), spider systematist and mentor, whose revisions were essential to my learning spider taxonomy.

**Conflicts of Interest:** The author declares no conflict of interest. The funders designed the collection of samples; my work was a part of a larger post-fire monitoring effort. The funders did not have a role in my data preparation or analysis; or in the preparation or writing of the manuscript; or the decision to publish the results.

## Appendix A

**Table A1.** Species and numbers collected for Ponderosa Pine Forest at Valles Caldera National Preserve 2011–2015. B = burned sites, U = unburned sites. Undet. = undetermined taxa.

| Order Family | Genus Species | 2011 | | 2012 | | 2013 | | 2014 | | 2015 | |
|---|---|---|---|---|---|---|---|---|---|---|---|
| | | B | U | B | U | B | U | B | U | B | U |
| Araneae Anyphaenidae | *Anyphaena marginalis* (Banks) | 0 | 135 | 1 | 50 | 0 | 123 | 0 | 112 | 0 | 133 |
| Clubionidae | *Clubiona oteroana* Gertsch | 5 | 3 | 0 | 0 | 0 | 0 | 1 | 0 | 2 | 0 |
| Dictynidae | *Cicurina* sp. | 21 | 7 | 25 | 9 | 34 | 19 | 10 | 14 | 30 | 14 |
| | *Hackmania saphes* (Chamberlin) | 0 | 0 | 0 | 0 | 0 | 6 | 0 | 2 | 0 | 5 |
| Gnaphosidae | *Drassodes neglectus* (Keyserling) | 7 | 5 | 8 | 31 | 2 | 15 | 7 | 5 | 1 | 1 |
| | *Gnaphosa* immatures | 17 | 19 | 11 | 49 | 13 | 17 | 5 | 12 | 3 | 23 |
| | *Gnaphosa muscorum* (L. Koch) | 6 | 65 | 23 | 192 | 20 | 152 | 56 | 103 | 21 | 43 |
| | *Gnaphosa parvula* Banks | 0 | 0 | 0 | 0 | 1 | 0 | 0 | 0 | 0 | 0 |
| | *Haplodrassus* immatures | 105 | 26 | 47 | 17 | 10 | 16 | 10 | 5 | 5 | 7 |
| | *Haplodrassus bicornis* (Emerton) | 0 | 0 | 0 | 0 | 1 | 1 | 0 | 0 | 0 | 0 |
| | *Haplodrassus eunis* Chamberlin | 0 | 0 | 1 | 16 | 0 | 26 | 3 | 31 | 1 | 24 |
| | *Haplodrassus signifier* (C.L. Koch) | 11 | 16 | 21 | 23 | 26 | 6 | 12 | 8 | 14 | 7 |
| | *Micaria* immatures | 0 | 0 | 1 | 1 | 2 | 1 | 2 | 0 | 3 | 0 |
| | *Micaria aenea* Thorell | 1 | 1 | 1 | 1 | 1 | 1 | 2 | 0 | 0 | 0 |
| | *Micaria foxi* Gertsch | 0 | 0 | 1 | 0 | 0 | 0 | 0 | 0 | 0 | 0 |
| | *Micaria gertschi* Barrows & Ivie | 0 | 0 | 0 | 0 | 2 | 0 | 1 | 0 | 0 | 0 |

**Table A1.** *Cont.*

| Order<br>Family | Genus<br>Species | 2011 B | 2011 U | 2012 B | 2012 U | 2013 B | 2013 U | 2014 B | 2014 U | 2015 B | 2015 U |
|---|---|---|---|---|---|---|---|---|---|---|---|
| | *Micaria pulicaria* (Sundevall) | 2 | 0 | 0 | 1 | 2 | 0 | 1 | 0 | 1 | 0 |
| | *Micaria riggsi* Gertsch | 0 | 0 | 5 | 0 | 5 | 0 | 1 | 0 | 2 | 0 |
| | *Zelotes* immatures | 16 | 7 | 2 | 13 | 1 | 4 | 0 | 5 | 1 | 4 |
| | *Zelotes fratris* Chamberlin | 17 | 9 | 3 | 10 | 3 | 17 | 0 | 15 | 4 | 9 |
| | *Zelotes lasalanus* Chamberlin | 0 | 0 | 1 | 2 | 2 | 0 | 2 | 0 | 0 | 0 |
| | *Zelotes puritanus* Chamberlin | 0 | 0 | 3 | 2 | 0 | 1 | 0 | 1 | 0 | 0 |
| Hahniidae | *Neoantistea gosiuta* Gertsch | 13 | 1 | 2 | 4 | 2 | 1 | 1 | 0 | 1 | 0 |
| Linyphiidae | *Agyneta* immatures | 3 | 3 | 0 | 0 | 0 | 0 | 0 | 0 | 0 | 0 |
| | *Agyneta simplex* (Emerton) | 6 | 0 | 0 | 0 | 0 | 0 | 0 | 0 | 0 | 0 |
| | *Agyneta uta* (Chamberlin) | 0 | 0 | 6 | 40 | 22 | 9 | 72 | 19 | 35 | 48 |
| | *Erigone dentosa* O. Pickard- Cambridge | 0 | 0 | 8 | 1 | 148 | 0 | 513 | 3 | 253 | 0 |
| | *Grammonota gentilis* Banks | 2 | 1 | 9 | 1 | 15 | 1 | 22 | 0 | 67 | 0 |
| | *Helophora orinoma* (Chamberlin) | 0 | 4 | 0 | 3 | 1 | 1 | 4 | 0 | 0 | 0 |
| | *Incestophantes lamprus* (Chamberlin) | 0 | 3 | 0 | 1 | 0 | 9 | 0 | 6 | 2 | 5 |
| | *Islandiana* immatures/undet. females | 0 | 0 | 2 | 0 | 2 | 0 | 0 | 0 | 0 | 0 |
| | *Islandiana coconino* Ivie | 0 | 0 | 0 | 0 | 0 | 0 | 2 | 0 | 0 | 0 |
| | *Islandiana flaveola* (Banks) | 35 | 0 | 15 | 4 | 15 | 0 | 7 | 0 | 13 | 0 |
| | *Islandiana lasalana* (Chamberlin & Ivie) | 0 | 0 | 0 | 0 | 0 | 0 | 0 | 1 | 1 | 0 |
| | *Lepthyphantes* immatures | 1 | 1 | 0 | 0 | 0 | 0 | 0 | 0 | 0 | 1 |
| | *Lepthyphantes intricatus* (Emerton) | 0 | 0 | 0 | 0 | 0 | 0 | 1 | 1 | 0 | 0 |
| | *Lepthyphantes turbatrix* (O. P.-Cambridge) | 0 | 0 | 0 | 3 | 0 | 0 | 0 | 1 | 0 | 3 |
| | *Mermessus* immatures | 6 | 0 | 7 | 1 | 4 | 3 | 5 | 2 | 2 | 1 |
| | *Mermessus major* Millidge | 0 | 0 | 0 | 0 | 0 | 0 | 1 | 0 | 0 | 0 |
| | *Mermessus taibo* (Chamberlin & Ivie) | 6 | 14 | 9 | 4 | 16 | 1 | 3 | 6 | 4 | 6 |
| | *Mermessus trilobatus* (Emerton) | 0 | 0 | 0 | 0 | 0 | 0 | 0 | 0 | 2 | 0 |
| | *Pocadicnemis occidentalis* Millidge | 0 | 9 | 1 | 75 | 0 | 31 | 0 | 77 | 0 | 75 |
| | *Spirembolus* immatures | 1 | 4 | 0 | 0 | 1 | 0 | 1 | 0 | 0 | 0 |
| | *Spirembolus pallidus* Chamberlin & Ivie | 0 | 4 | 0 | 7 | 0 | 9 | 4 | 31 | 2 | 18 |
| | *Spirembolus spirotubus* (Banks) | 1 | 0 | 0 | 0 | 1 | 0 | 0 | 0 | 0 | 0 |
| | *Tachygyna* immatures | 0 | 0 | 0 | 0 | 0 | 0 | 0 | 0 | 0 | 5 |
| | *Tachygyna haydeni* Chamberlin & Ivie | 0 | 1 | 0 | 1 | 0 | 0 | 1 | 2 | 1 | 0 |
| | *Tachygyna tuoba* Chamberlin & Ivie | 1 | 4 | 0 | 13 | 0 | 11 | 1 | 33 | 0 | 8 |
| | *Walckenaeria* immatures | 1 | 0 | 0 | 1 | 0 | 0 | 0 | 0 | 0 | 2 |
| | *Walckenaeria communis* (Emerton) | 0 | 0 | 0 | 2 | 0 | 0 | 0 | 1 | 0 | 3 |
| | *Walckenaeria maesta* Millidge | 0 | 0 | 0 | 2 | 0 | 4 | 0 | 0 | 0 | 0 |
| | *Walckenaeria spiralis* (Emerton) | 0 | 5 | 0 | 4 | 0 | 41 | 0 | 12 | 0 | 28 |
| | *Wubana drassoides* (Emerton) | 0 | 9 | 0 | 13 | 0 | 8 | 1 | 8 | 0 | 10 |
| Lycosidae | *Alopecosa kochi* (Keyserling) | 15 | 46 | 3 | 141 | 0 | 31 | 17 | 55 | 7 | 33 |
| | *Hogna* sp. | 1 | 12 | 0 | 19 | 1 | 7 | 0 | 2 | 0 | 0 |
| | *Pardosa* immatures | 8 | 6 | 12 | 81 | 7 | 171 | 45 | 23 | 180 | 40 |
| | *Pardosa coloradensis* Banks | 0 | 0 | 0 | 0 | 10 | 0 | 11 | 0 | 52 | 0 |
| | *Pardosa concinna* (Thorell) | 0 | 0 | 11 | 1 | 97 | 1 | 137 | 2 | 142 | 0 |
| | *Pardosa distincta* (Blackwall) | 0 | 0 | 0 | 0 | 36 | 1 | 172 | 1 | 363 | 0 |
| | *Pardosa montgomeryi* Gertsch | 0 | 11 | 0 | 0 | 1 | 0 | 14 | 0 | 68 | 0 |
| | *Pardosa uncata* (Thorell) | 20 | 21 | 13 | 112 | 14 | 43 | 4 | 62 | 6 | 115 |
| | *Pardosa xerophila* Vogel | 0 | 0 | 0 | 0 | 9 | 0 | 31 | 0 | 23 | 1 |
| | *Pardosa yavapa* Chamberlin | 0 | 11 | 1 | 142 | 1 | 51 | 13 | 84 | 2 | 150 |
| Philodromidae | *Thanatus* immatures | 0 | 1 | 1 | 0 | 0 | 0 | 0 | 0 | 1 | 0 |
| | *Thanatus formicinus* (Clerck) | 0 | 0 | 1 | 1 | 0 | 0 | 0 | 0 | 0 | 0 |
| | *Thanatus coloradensis* Keyserling | 0 | 0 | 1 | 1 | 0 | 0 | 2 | 0 | 1 | 1 |
| | Thanatus vulgaris Simon | 0 | 0 | 0 | 0 | 0 | 0 | 1 | 0 | 0 | 0 |
| Phrurolithidae | *Scotinella pugnata* (Emerton) | 8 | 0 | 2 | 0 | 0 | 0 | 0 | 0 | 0 | 0 |
| Thomisidae | *Xysticus* immatures | 9 | 5 | 5 | 16 | 7 | 13 | 3 | 8 | 1 | 5 |
| | *Xysticus apachecus* Gertsch | 0 | 3 | 0 | 1 | 0 | 1 | 0 | 0 | 0 | 1 |
| | *Xysticus cunctator* Thorell | 0 | 0 | 0 | 0 | 4 | 0 | 1 | 0 | 0 | 0 |
| | *Xysticus emertoni* Keyserling | 0 | 0 | 0 | 8 | 0 | 5 | 2 | 6 | 4 | 7 |
| | *Xysticus ferox* (Hentz) | 0 | 0 | 0 | 0 | 0 | 0 | 3 | 0 | 3 | 0 |
| | *Xysticus gulosus* Keyserling | 0 | 0 | 0 | 0 | 0 | 0 | 0 | 1 | 1 | 0 |
| | *Xysticus locuples* Keyserling | 0 | 0 | 0 | 1 | 0 | 0 | 0 | 0 | 0 | 0 |
| | *Xysticus luctuosus* (Blackwall) | 0 | 0 | 0 | 0 | 0 | 0 | 1 | 0 | 0 | 0 |
| | *Xysticus montanensis* Keyserling | 1 | 0 | 1 | 3 | 0 | 8 | 3 | 1 | 0 | 2 |
| Opiliones<br>Sclerosomatidae | *Leiobunum* sp. | 2 | 0 | 1 | 15 | 6 | 92 | 2 | 0 | 6 | 38 |
| | *Togwoteeus biceps* (Thorell) | 4 | 62 | 55 | 9 | 28 | 93 | 47 | 7 | 45 | 158 |
| Acari<br>Mesostigmata | | 1 | 0 | 3 | 34 | 64 | 1 | 25 | 1 | 468 | 156 |
| Oribatida | | 2 | 1 | 1 | 15 | 1 | 6 | 0 | 15 | 0 | 9 |
| Trombidiformes | Undetermined | 0 | 4 | 4 | 0 | 0 | 6 | 3 | 27 | 3 | 8 |
| Anystidae | | 0 | 6 | 0 | 34 | 2 | 4 | 0 | 16 | 1 | 9 |
| Erythraeidae | | 27 | 28 | 182 | 20 | 183 | 43 | 96 | 8 | 326 | 2 |

**Table A2.** Species and numbers collected for Mixed-conifer Forest at Valles Caldera National Preserve 2011–2015. B = burned sites, U = unburned sites. Undet. = undetermined taxa.

| Order<br>Family | Genus<br>Species | 2011 B | 2011 U | 2012 B | 2012 U | 2013 B | 2013 U | 2014 B | 2014 U | 2015 B | 2015 U |
|---|---|---|---|---|---|---|---|---|---|---|---|
| Araneae<br>Amaurobiidae | *Callobius arizonicus* (Chamberlin & Ivie) | 0 | 5 | 1 | 12 | 0 | 5 | 0 | 3 | 0 | 1 |
| Dictynidae | *Cicurina* sp. | 22 | 58 | 53 | 55 | 65 | 39 | 211 | 36 | 99 | 57 |
| Gnaphosidae | *Drassodes neglectus* (Keyserling) | 9 | 5 | 19 | 14 | 18 | 7 | 4 | 7 | 8 | 1 |
| | *Gnaphosa* immatures | 17 | 7 | 21 | 13 | 66 | 24 | 9 | 8 | 72 | 10 |
| | *Gnaphosa muscorum* (L. Koch) | 3 | 8 | 71 | 73 | 40 | 56 | 137 | 71 | 57 | 45 |
| | *Gnaphosa parvula* Banks | 0 | 1 | 2 | 0 | 0 | 0 | 2 | 0 | 2 | 0 |
| | *Haplodrassus* immatures | 64 | 8 | 13 | 22 | 11 | 21 | 9 | 25 | 11 | 13 |
| | *Haplodrasus bicornis* (Emerton) | 0 | 0 | 0 | 0 | 1 | 0 | 0 | 0 | 0 | 0 |
| | *Haplodrassus eunis* Chamberlin | 0 | 0 | 9 | 15 | 8 | 6 | 27 | 12 | 8 | 14 |
| | *Haplodrassus signifier* (C.L. Koch) | 29 | 7 | 25 | 20 | 11 | 9 | 41 | 21 | 23 | 12 |
| | *Micaria* immatures | 0 | 0 | 2 | 2 | 6 | 2 | 2 | 1 | 1 | 0 |
| | *Micaria aenea* Thorell | 0 | 0 | 0 | 10 | 0 | 5 | 0 | 0 | 0 | 0 |
| | *Micaria gertschi* Barrows & Ivie | 0 | 0 | 0 | 3 | 6 | 7 | 1 | 0 | 1 | 1 |
| | *Micaria pulicaria* (Sundevall) | 0 | 1 | 1 | 5 | 1 | 0 | 1 | 0 | 1 | 1 |
| | *Micaria riggsi* Gertsch | 0 | 0 | 0 | 0 | 4 | 0 | 3 | 0 | 4 | 0 |
| | *Micaria rossica* Thorell | 0 | 0 | 1 | 0 | 8 | 0 | 32 | 0 | 44 | 1 |
| | *Orodrassus coloradensis* (Emerton) | 0 | 0 | 1 | 4 | 0 | 4 | 0 | 1 | 0 | 0 |
| | *Zelotes* immatures | 8 | 5 | 5 | 7 | 0 | 12 | 3 | 5 | 1 | 3 |
| | *Zelotes fratris* Chamberlin | 7 | 4 | 1 | 10 | 0 | 4 | 7 | 3 | 5 | 3 |
| | *Zelotes puritanus* Chamberlin | 1 | 0 | 5 | 0 | 11 | 0 | 5 | 1 | 4 | 0 |
| Hahniidae | *Neoantistea gosiuta* Gertsch | 4 | 1 | 0 | 2 | 1 | 4 | 1 | 0 | 2 | 1 |
| Linyphiidae | *Agyneta* immatures/undet. females | 0 | 0 | 2 | 4 | 8 | 6 | 4 | 1 | 0 | 2 |
| | *Agyneta danielbelangeri* Duperre | 0 | 0 | 0 | 0 | 0 | 0 | 0 | 0 | 1 | 13 |
| | *Agyneta uta* (Chamberlin) | 0 | 0 | 0 | 2 | 3 | 1 | 52 | 7 | 7 | 35 |
| | *Ceratinella brunnea* Emerton | 2 | 1 | 0 | 3 | 0 | 2 | 0 | 1 | 0 | 2 |
| | *Ceratinella ornatula* (Crosby & Bishop) | 0 | 0 | 0 | 0 | 0 | 8 | 0 | 5 | 3 | 0 |
| | *Erigone dentosa* O. Pickard-Cambridge | 1 | 5 | 3 | 10 | 43 | 0 | 403 | 1 | 324 | 25 |
| | *Grammonota gentilis* Banks | 0 | 1 | 0 | 1 | 4 | 0 | 1 | 0 | 24 | 1 |
| | *Helophora orinoma* (Chamberlin) | 0 | 18 | 0 | 34 | 1 | 28 | 2 | 39 | 3 | 70 |
| | *Incestophantes lamprus* (Chamberlin) | 0 | 0 | 0 | 2 | 3 | 2 | 0 | 0 | 0 | 3 |
| | *Islandiana coconino* Ivie | 0 | 0 | 0 | 0 | 3 | 0 | 0 | 0 | 1 | 0 |
| | *Islandiana flaveola* (Banks) | 0 | 0 | 0 | 0 | 2 | 1 | 0 | 1 | 0 | 6 |
| | *Lepthyphantes* immatures | 3 | 1 | 0 | 3 | 1 | 1 | 0 | 0 | 0 | 1 |
| | *Lepthyphantes intricatus* (Emerton) | 0 | 1 | 1 | 1 | 2 | 0 | 5 | 0 | 0 | 0 |
| | *Lepthyphantes turbatrix* (O. Pickard-Cambridge) | 0 | 0 | 0 | 1 | 3 | 0 | 4 | 2 | 5 | 1 |
| | *Mermessus* immatures/undet. females | 17 | 6 | 4 | 0 | 3 | 0 | 6 | 0 | 3 | 0 |
| | *Mermessus taibo* (Chamberlin & Ivie) | 1 | 0 | 11 | 4 | 18 | 1 | 10 | 0 | 14 | 0 |
| | *Mermessus trilobatus* (Emerton) | 0 | 0 | 9 | 0 | 0 | 0 | 0 | 0 | 3 | 0 |
| | *Pocadicnemis occidentalis* Millidge | 0 | 2 | 1 | 3 | 0 | 3 | 0 | 2 | 0 | 0 |
| | *Scotinotylus* undet. females | 0 | 1 | 1 | 3 | 0 | 0 | 3 | 0 | 0 | 2 |
| | *Scotinotylus pallidus* (Emerton) | 0 | 0 | 0 | 1 | 0 | 0 | 0 | 0 | 0 | 4 |
| | *Scotinotylus pollucis* Millidge | 1 | 0 | 0 | 0 | 0 | 1 | 0 | 4 | 0 | 0 |
| | *Scotinotylus sanctus* (Crosby) | 5 | 1 | 0 | 0 | 0 | 0 | 0 | 0 | 0 | 0 |
| | *Sisicottus* immatures/undet. females | 0 | 0 | 0 | 1 | 0 | 0 | 0 | 0 | 0 | 2 |
| | *Sisicottus montanus* (Emerton) | 0 | 5 | 0 | 29 | 0 | 12 | 0 | 8 | 0 | 0 |
| | *Sisicottus orites* (Chamberlin) | 0 | 1 | 0 | 1 | 0 | 0 | 1 | 53 | 0 | 43 |
| | *Spirembolus* immatures/undet. females | 0 | 2 | 0 | 1 | 0 | 0 | 0 | 0 | 0 | 0 |
| | *Spirembolus pallidus* Chamberlin & Ivie | 0 | 0 | 0 | 1 | 1 | 0 | 0 | 6 | 0 | 1 |
| | *Spirembolus spirotubus* (Banks) | 0 | 0 | 0 | 1 | 0 | 0 | 0 | 0 | 0 | 0 |
| | *Tachygyna* immatures/undet. females | 0 | 0 | 0 | 0 | 0 | 1 | 1 | 5 | 0 | 4 |
| | *Tachygyna haydeni* Chamberlin & Ivie | 0 | 1 | 0 | 7 | 0 | 4 | 0 | 10 | 0 | 30 |
| | *Tachygyna tuoba* Chamberlin & Ivie | 0 | 1 | 0 | 2 | 0 | 20 | 0 | 30 | 0 | 6 |
| | *Tapinocyba* sp. | 0 | 0 | 0 | 0 | 1 | 6 | 0 | 4 | 2 | 23 |
| | *Tapinocyba* cf. *cameroni* Duperre & Paquin | 0 | 0 | 0 | 0 | 0 | 0 | 0 | 0 | 0 | 1 |
| | *Tapinocyba minuta* (Emerton) | 0 | 0 | 0 | 1 | 0 | 0 | 0 | 0 | 0 | 0 |
| | *Walckenaeria communis* (Emerton) | 2 | 1 | 0 | 2 | 0 | 0 | 0 | 2 | 0 | 3 |
| | *Wubana drassoides* (Emerton) | 0 | 26 | 0 | 8 | 0 | 8 | 0 | 12 | 0 | 17 |
| Lycosidae | *Alopecosa kochi* (Keyserling) | 4 | 0 | 2 | 27 | 9 | 7 | 12 | 8 | 12 | 10 |
| | *Hogna* sp. | 0 | 4 | 0 | 0 | 0 | 3 | 2 | 0 | 3 | 4 |
| | *Pardosa* immatures | 10 | 15 | 4 | 55 | 15 | 113 | 18 | 23 | 94 | 24 |
| | *Pardosa coloradensis* Banks | 0 | 0 | 1 | 0 | 0 | 0 | 8 | 0 | 94 | 1 |
| | *Pardosa concinna* (Thorell) | 0 | 0 | 9 | 0 | 22 | 0 | 196 | 1 | 476 | 1 |
| | *Pardosa distincta* (Blackwall) | 0 | 1 | 0 | 3 | 5 | 1 | 36 | 1 | 163 | 6 |
| | *Pardosa montgomeryi* Gertsch | 0 | 0 | 0 | 3 | 5 | 0 | 13 | 0 | 49 | 0 |

**Table A2.** *Cont.*

| Order<br>Family | Genus<br>Species | 2011 B | 2011 U | 2012 B | 2012 U | 2013 B | 2013 U | 2014 B | 2014 U | 2015 B | 2015 U |
|---|---|---|---|---|---|---|---|---|---|---|---|
| | *Pardosa uncata* (Thorell) | 26 | 43 | 42 | 280 | 69 | 111 | 199 | 120 | 123 | 97 |
| | *Pardosa xerophila* Vogel | 0 | 0 | 0 | 0 | 0 | 0 | 10 | 1 | 0 | 2 |
| | *Pardosa yavapa* Chamberlin | 0 | 0 | 0 | 0 | 0 | 0 | 11 | 0 | 5 | 1 |
| | *Trochosa terricola* Thorell | 0 | 0 | 0 | 5 | 0 | 1 | 0 | 1 | 0 | 3 |
| Phrurolithidae | *Phrurolithus camawhitae* Gertsch | 1 | 0 | 17 | 2 | 0 | 0 | 0 | 0 | 0 | 0 |
| | *Phrurolithus connectus* Gertsch | 1 | 0 | 0 | 0 | 0 | 0 | 0 | 0 | 0 | 0 |
| | *Phrurolithus schwarzi* Gertsch | 1 | 0 | 0 | 0 | 0 | 0 | 0 | 0 | 0 | 0 |
| Salticidae | *Pelegrina flavipes* (Peckham & Peckham) | 0 | 0 | 0 | 0 | 0 | 10 | 0 | 1 | 0 | 0 |
| Theridiidae | *Steatoda* immatues | 0 | 0 | 0 | 4 | 0 | 1 | 2 | 4 | 0 | 0 |
| | *Steatoda albomaculata* (DeGeer) | 0 | 0 | 0 | 0 | 1 | 0 | 0 | 0 | 0 | 0 |
| | *Steatoda hespera* Chamberlin & Ivie | 0 | 0 | 0 | 8 | 0 | 0 | 0 | 0 | 0 | 3 |
| Thomisidae | *Xysticus* immatures | 4 | 4 | 5 | 3 | 11 | 8 | 6 | 3 | 6 | 0 |
| | *Xysticus cunctator* Thorell | 0 | 0 | 1 | 0 | 1 | 0 | 1 | 0 | 9 | 0 |
| | *Xysticus emertoni* Keyserling | 0 | 0 | 0 | 0 | 0 | 1 | 1 | 0 | 8 | 0 |
| | *Xysticus ferox* (Hentz) | 0 | 0 | 0 | 0 | 0 | 1 | 0 | 0 | 0 | 0 |
| | *Xysticus locuples* Keyserling | 0 | 1 | 0 | 0 | 0 | 0 | 0 | 0 | 0 | 0 |
| | *Xysticus luctuosus* Keyserling | 0 | 0 | 0 | 0 | 0 | 0 | 0 | 0 | 0 | 1 |
| | *Xysticus montanensis* Keyserling | 0 | 2 | 0 | 1 | 1 | 4 | 0 | 0 | 0 | 1 |
| | *Xysticus triguttatus* Keyserling | 0 | 0 | 0 | 0 | 0 | 0 | 1 | 0 | 0 | 0 |
| Opiliones<br>Paronychidae | *Sclerobunus robustus* (Packard) | 5 | 0 | 2 | 5 | 1 | 2 | 1 | 4 | 0 | 0 |
| Sclerosomatidae | *Leiobunum* sp. | 0 | 0 | 23 | 1716 | 46 | 1259 | 30 | 139 | 51 | 673 |
| | *Togwoteeus biceps* (Thorell) | 2 | 0 | 83 | 978 | 113 | 721 | 237 | 430 | 296 | 1102 |
| Acari<br>Mesostigmata | | 0 | 0 | 0 | 16 | 16 | 1 | 18 | 8 | 440 | 16 |
| Oribatida | | 0 | 0 | 0 | 81 | 0 | 195 | 0 | 302 | 4 | 76 |
| Trombidiformes | Undetermined | 0 | 0 | 0 | 14 | 1 | 1 | 0 | 14 | 0 | 0 |
| Anystidae | | 0 | 0 | 0 | 10 | 1 | 10 | 0 | 32 | 0 | 6 |
| Bdellidae | | 0 | 0 | 0 | 6 | 0 | 6 | 0 | 7 | 0 | 3 |
| Erythraeidae | | 0 | 0 | 77 | 120 | 181 | 75 | 108 | 57 | 2 | 42 |

**Table A3.** Species and numbers collected for Mountain Valley Grassland at Valles Caldera National Preserve 2011–2015. B = burned sites, U = unburned sites. Undet. = undetermined taxa.

| Order<br>Family | Genus<br>Species | 2011 B | 2011 U | 2012 B | 2012 U | 2013 B | 2013 U | 2014 B | 2014 U | 2015 B | 2015 U |
|---|---|---|---|---|---|---|---|---|---|---|---|
| Araneae<br>Clubionidae | *Clubiona oteroana* Gertsch | 32 | 13 | 12 | 14 | 2 | 6 | 7 | 9 | 19 | 26 |
| Dictynidae | *Cicurina* sp. | 1 | 6 | 0 | 2 | 2 | 6 | 0 | 3 | 0 | 4 |
| Gnaphosidae | *Gnaphosa* immatures | 0 | 13 | 2 | 16 | 1 | 34 | 0 | 78 | 2 | 34 |
| | *Gnaphosa borea* Kulczynski | 0 | 0 | 0 | 0 | 0 | 1 | 0 | 0 | 0 | 0 |
| | *Gnaphosa muscorum* (L. Koch) | 2 | 0 | 5 | 3 | 5 | 0 | 5 | 1 | 5 | 0 |
| | *Gnaphosa parvula* Banks | 4 | 50 | 0 | 31 | 0 | 44 | 0 | 178 | 1 | 59 |
| | *Haplodrassus signifier* (C.L. Koch) | 327 | 116 | 239 | 209 | 177 | 170 | 170 | 210 | 103 | 88 |
| | *Micaria* immatures | 12 | 3 | 0 | 2 | 10 | 3 | 4 | 0 | 1 | 0 |
| | *Micaria aenea* Thorell | 0 | 0 | 0 | 0 | 5 | 1 | 0 | 0 | 0 | 0 |
| | *Micaria gertschi* Barrows & Ivie | 10 | 1 | 0 | 0 | 0 | 0 | 0 | 0 | 1 | 1 |
| | *Micaria riggsi* Gertsch | 0 | 0 | 3 | 0 | 0 | 2 | 2 | 0 | 1 | 0 |
| | *Micaria rossica* Thorell | 1 | 0 | 26 | 3 | 93 | 11 | 79 | 4 | 41 | 0 |
| | *Zelotes* immatures | 6 | 2 | 11 | 0 | 3 | 2 | 2 | 6 | 0 | 2 |
| | *Zelotes fratris* Chamberlin | 0 | 0 | 0 | 0 | 0 | 0 | 0 | 0 | 2 | 0 |
| | *Zelotes lasalanus* Chamberlin | 3 | 0 | 3 | 15 | 4 | 18 | 3 | 23 | 7 | 8 |
| Hahniidae | *Neoantistea gosiuta* Gertsch | 16 | 19 | 6 | 15 | 7 | 18 | 4 | 48 | 7 | 19 |
| Linyphiidae | *Agyneta* immatures | 31 | 15 | 8 | 12 | 22 | 15 | 24 | 12 | 13 | 14 |
| | *Agyneta hedini* Paquin & Duperre | 0 | 0 | 0 | 0 | 0 | 0 | 0 | 0 | 1 | 0 |
| | *Agyneta simplex* (Emerson) | 15 | 1 | 0 | 0 | 0 | 0 | 0 | 0 | 0 | 0 |
| | *Agyneta uta* (Chamberlin) | 0 | 0 | 9 | 18 | 11 | 14 | 59 | 36 | 34 | 16 |
| | *Ceratinella brunnea* Emerton | 6 | 2 | 1 | 2 | 10 | 12 | 4 | 6 | 3 | 1 |
| | *Erigone dentosa* O. Pickard-Cambridge | 6 | 2 | 1 | 15 | 4 | 20 | 88 | 23 | 212 | 110 |

**Table A3.** *Cont.*

| Order Family | Genus Species | 2011 B | 2011 U | 2012 B | 2012 U | 2013 B | 2013 U | 2014 B | 2014 U | 2015 B | 2015 U |
|---|---|---|---|---|---|---|---|---|---|---|---|
| | *Grammonota gentilis* Banks | 40 | 130 | 95 | 189 | 26 | 116 | 18 | 169 | 63 | 273 |
| | *Islandiana* immatures | 0 | 0 | 0 | 0 | 6 | 7 | 8 | 20 | 9 | 30 |
| | *Islandiana coconino* Ivie | 18 | 1 | 8 | 0 | 27 | 4 | 17 | 1 | 33 | 3 |
| | *Islandiana flaveola* (Banks) | 14 | 14 | 18 | 18 | 28 | 32 | 37 | 43 | 15 | 12 |
| | *Mermessus* immatures/undet. females | 0 | 2 | 0 | 5 | 1 | 0 | 2 | 6 | 0 | 2 |
| | *Mermessus major* (Millidge) | 0 | 1 | 0 | 0 | 0 | 0 | 0 | 0 | 0 | 1 |
| | *Mermessus taibo* (Chamberlin & Ivie) | 0 | 1 | 0 | 0 | 0 | 1 | 3 | 2 | 1 | 0 |
| | *Mermessus trilobatus* (Emerton) | 7 | 11 | 3 | 0 | 1 | 2 | 0 | 4 | 7 | 7 |
| | *Spirembolus pallidus* Chamberlin & Ivie | 0 | 0 | 0 | 0 | 0 | 0 | 2 | 3 | 0 | 1 |
| | *Spirembolus spirotubus* (Banks) | 0 | 0 | 3 | 3 | 5 | 12 | 0 | 0 | 0 | 0 |
| | *Tachygyna haydeni* Chamberlin & Ivie | 0 | 0 | 8 | 0 | 34 | 5 | 0 | 0 | 0 | 0 |
| | *Tachygyna tuoba* Chamberlin & Ivie | 0 | 0 | 1 | 3 | 0 | 0 | 0 | 0 | 0 | 0 |
| | *Tapinocyba* sp. | 0 | 0 | 0 | 0 | 0 | 0 | 30 | 0 | 0 | 0 |
| | *Tapinocyba dietrichi* Crosby & Bishop | 0 | 0 | 0 | 0 | 0 | 8 | 0 | 6 | 0 | 1 |
| | *Walckenaeria* immatures | 0 | 2 | 0 | 2 | 0 | 5 | 0 | 15 | 1 | 4 |
| | *Walckenaeria communis* (Emerton) | 0 | 0 | 0 | 1 | 0 | 4 | 0 | 5 | 0 | 0 |
| | *Walckenaeria dondalei* Millidge | 0 | 0 | 0 | 0 | 0 | 1 | 0 | 2 | 0 | 1 |
| | *Walckenaeria spiralis* (Emerton) | 0 | 1 | 0 | 0 | 0 | 0 | 0 | 0 | 0 | 0 |
| Lycosidae | *Alopecosa kochi* (Keyserling) | 6 | 1 | 6 | 1 | 5 | 5 | 10 | 2 | 11 | 6 |
| | *Hogna* sp. | 0 | 0 | 1 | 14 | 4 | 4 | 8 | 10 | 7 | 8 |
| | *Pardosa* immatures | 97 | 58 | 36 | 78 | 64 | 100 | 108 | 129 | 67 | 62 |
| | *Pardosa coloradensis* Banks | 0 | 0 | 0 | 6 | 9 | 37 | 10 | 9 | 4 | 14 |
| | *Pardosa concinna* (Thorell) | 100 | 190 | 392 | 595 | 218 | 214 | 449 | 368 | 416 | 367 |
| | *Pardosa distincta* (Blackwall) | 327 | 568 | 483 | 1002 | 439 | 798 | 893 | 1408 | 777 | 1340 |
| | *Pardosa montgomeryi* Gertsch | 32 | 15 | 39 | 8 | 102 | 5 | 166 | 12 | 170 | 36 |
| | *Pardosa xerophila* Vogel | 16 | 8 | 0 | 0 | 0 | 2 | 15 | 1 | 38 | 0 |
| | *Schizocosa mccooki* (Montgomery) | 1 | 3 | 0 | 0 | 4 | 3 | 0 | 0 | 0 | 0 |
| Philodromidae | *Ebo* immatures | 0 | 1 | 1 | 11 | 1 | 6 | 0 | 0 | 4 | 13 |
| | *Ebo pepinensis* Gertsch | 3 | 1 | 0 | 0 | 0 | 4 | 0 | 0 | 1 | 11 |
| | *Ebo punctatus* Sauer & Platnick | 0 | 0 | 0 | 0 | 0 | 0 | 5 | 12 | 0 | 0 |
| | *Thanatus* immatures | 79 | 22 | 21 | 27 | 28 | 37 | 16 | 29 | 18 | 9 |
| | *Thanatus coloradensis* Keyserling | 48 | 14 | 33 | 42 | 89 | 75 | 63 | 31 | 27 | 18 |
| | *Thanatus formicinus* (Clerck) | 0 | 0 | 1 | 0 | 0 | 1 | 0 | 0 | 0 | 0 |
| | *Thanatus vulgaris* Simon | 0 | 0 | 0 | 11 | 0 | 44 | 0 | 3 | 2 | 4 |
| Salticidae | *Pellenes* sp. | 3 | 4 | 0 | 1 | 1 | 4 | 0 | 0 | 1 | 1 |
| | *Phidippus olympus* Edwards | 6 | 0 | 6 | 4 | 3 | 2 | 0 | 0 | 1 | 0 |
| Theridiidae | *Euryopis* immatures | 1 | 1 | 0 | 0 | 0 | 0 | 0 | 0 | 0 | 0 |
| | *Euryopis saukea* Levi | 1 | 2 | 0 | 0 | 0 | 8 | 0 | 0 | 0 | 3 |
| | *Euryopis scriptipes* Banks | 0 | 0 | 0 | 0 | 0 | 1 | 0 | 0 | 0 | 0 |
| Thomisidae | *Xysticus* immatures | 36 | 58 | 61 | 40 | 59 | 80 | 50 | 49 | 60 | 49 |
| | *Xysticus apachecus* Gertsch | 1 | 0 | 2 | 0 | 0 | 0 | 0 | 0 | 0 | 0 |
| | *Xysticus cunctator* Thorell | 0 | 0 | 1 | 0 | 2 | 0 | 1 | 1 | 0 | 0 |
| | *Xysticus ellipticus* Turnbull, Dondale & Redner | 0 | 0 | 0 | 1 | 0 | 1 | 0 | 0 | 0 | 0 |
| | *Xysticus emertoni* Keyserling | 0 | 0 | 1 | 0 | 2 | 5 | 0 | 0 | 0 | 1 |
| | *Xysticus ferox* (Hentz) | 6 | 8 | 57 | 61 | 16 | 24 | 26 | 90 | 8 | 49 |
| | *Xysticus montanensis* Keyserling | 3 | 13 | 7 | 31 | 1 | 24 | 1 | 60 | 3 | 38 |
| | *Xysticus paiutus* Gertsch | 0 | 7 | 9 | 14 | 15 | 1 | 27 | 13 | 26 | 9 |
| | *Xysticus triguttatus* Keyserling | 102 | 12 | 67 | 57 | 70 | 32 | 124 | 43 | 33 | 13 |
| Opiliones Sclerosomatidae | *Leiobunum* sp. | 0 | 0 | 0 | 14 | 0 | 27 | 0 | 5 | 0 | 0 |
| | *Togwoteeus biceps* (Thorell) | 0 | 1 | 0 | 22 | 0 | 3 | 1 | 2 | 1 | 2 |
| Acari Mesostigmata | | 5 | 2 | 3 | 17 | 28 | 176 | 91 | 227 | 482 | 356 |
| Oribatida | | 1 | 0 | 1 | 0 | 2 | 5 | 1 | 3 | 0 | 0 |
| Trombidiformes Anystidae | | 32 | 10 | 0 | 2 | 10 | 15 | 1 | 0 | 17 | 9 |
| Erythraeidae | | 2170 | 1698 | 1269 | 382 | 2730 | 678 | 732 | 536 | 798 | 401 |

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
