# Peer review of "Responses of Ground-Dwelling Spider (Arachnida: Araneae) Communities to Wildfire in Three Habitats in Northern New Mexico, USA, with Notes on Mites and Harvestmen (Arachnida: Acari, Opiliones)"

_diversity, doi:10.3390/d12100396_

Round 1

Reviewer 1 Report

Review of “Responses of ground-dwelling spider (Arachnida: Araneae) communities to wildfire in three habitats in northern New Mexico, with notes on mites and harvestmen (Arachnida: Acari, Opiliones)” by Sandra Brantley

This is a manuscript that describes the results of a five-year study on the response of arachnids (mostly spiders) to fire in three New Mexican habitats: Ponderosa Pine habitat, Mixed-conifer habitat, and Mountain valley grassland habitat. The manuscript is written well, and the statistical analyses seem sound and well thought out.

There are specific comments in the attached PDF, but I will address some of the more critical comments here:

  • Brantley discusses the Thompson Ridge Fire as interrupting her study and manipulating her data so that abundance values are different among sites that experienced this unplanned fire versus those that did not. I wonder if, overall, data could be standardized better if they were divided by sample size. In this way, abundance data would not be affected as greatly by how many samples were collected at a particular location. Perhaps it would not completely mitigate the effect of this unplanned fire, but perhaps it could help?
  • NMDS plots look like they are made with the general R program but could be made clearer with the use of a plotting package in R such as ggplot2, which I use and recommend.
  • There are some parts of the results that include analyses of data – such as why linyphiids or Micaria were present in certain traps. I think these discussions should belong in the Discussion section.
  • Some of the captions need expanding to accurately describe the graphs or tables indicated.

Really, this manuscript is like three manuscripts in one since Brantley did not compare habitats, as they were completely different in species composition. I commend her on the amount of work that must’ve went into this manuscript and this study in general, as it seems like a large undertaking. Even though the paper seems to be a combination of analyses of these three habitats that are not directly compared, she does a good job of making it seem like one large study (which, in a way, it is) with combined tables (with all three habitats) and a single discussion that brings everything together at the end.

Overall, I feel that this is a solidly written manuscript that is a result of a large, well-conducted ecological study. I recommend it for publication given that these minor issues are resolved.

Author Response

Diversity manuscript reviews – Sandra Brantley’s responses are in italics

Reviewer 1

Line 46:  I removed the comma

Lines 124-126: “Comparison of what among sites? Abundance?”  Yes, relative abundance. I have corrected the text.

Line 157: Use ggplot2 for better NMDS graphs; NMDS for PP: change B and U to burned and unburned – I improved the legends and colors.

Line 165 so far I can’t correct, and Lines 305-307: Tables. The columns were in line in the version I submitted. They were in landscape format originally. I will work with the editor to re- format the tables.

Lines 215-216: “Why not divide by sample to create proportions to control for the asymmetry?” My intent was to show the variability in the samples, which is typical for arthropods; I thought the violin plots were more informative, although descriptive. I have added the significant adonis pairwise comparisons among years to the figure captions.

Lines 226-229: about Micaria, move to discussion. Also mentioned by another reviewer. I have removed the sentences from the Results.

Lines 290-291: “unclear what average similarity is comparing – add more info to table caption” The comparisons are meant to show that burned and unburned sites within each forest were less similar to each other (they were more severely burned) than the burned and unburned sites within the grassland habitat (which experienced only a light burn).

Lines 143-154 I have rewritten the paragraph: In PRIMER I used Similarity Percentages (SIMPER), which compared Bray-Curtis dissimilarities in the mean between-group differences in relative abundance among samples by treatment and among years. The method can be influenced by large variation within group. I also used this to show the percent contribution of the most abundant genera for treatment and year. Indicator genera (indicspp in R) showed taxa that were associated with a particular treatment or year.

For the table add: Burned and unburned treatments in each severely burned forest are not as similar to each other, compared to the greater similarity between treatments at the mountain valley grassland, which was less severely burned.

Line 448: area to areas  Corrected

Reviewer 2 Report

See attached file

Author Response

Diversity manuscript reviews – Sandra Brantley’s responses are in italics

Reviewer 2

Line 21: YearFis  - A typo, corrected to Years.

Line 28: is there a definition of drought, how severe? Yes, the Palmer Drought Severity Index tracks drought and wet periods based on temperature and precipitation, and standardized so that positive values are wet periods, negative values are droughts. I have added a few sentences about the drought conditions in summer in New Mexico for the years of the study. I also added the web page source of information to the literature cited.

Line 46: delete comma – noted by another reviewer as well; I have removed the comma

Materials and methods

71-73 thank you for including common and scientific names. I try to include both when I can, but many spiders do not have common names.

Fig 3 Add what MC and PP stand for to Figure legend. (OK in text as previously defined; but figures and tables need to be able to stand on their own.) I have added the full habitat names along with the abbreviations.

94 how were sites selected? Random? Ease of access? Typical appearance.  Looking at the site map, now Figure 2, you can see some sites that are close together, but they are on different slopes, for example, which separated ponderosa pine from mixed-conifer stands, or where forest and meadow were adjacent. There were limits to site selection based on the area covered by the fire, but we avoided habitat edges and sites were at least 300-500 m apart. I clarified the text.

112 …for analysis, which also…  comma inserted

126 adonis measures what? Same w 136 Pairwise comparisons in adonis were based on relative abundance.

Results

162 Pardosa is what? Pardosa is a common and widespread wolf spider, one of the few genera in this study with high relative abundance. In my other projects they seem to be everywhere, all the time. I have added the common family names for Pardosa, Haplodrassus and Gnaphosa in the Results. I hoped the Appendices would do two things: give the taxonomy of the groups so that it didn’t interfere with the text; and to give the list of taxa at the species level, if I could identify them. The reader can then see which species were collected, even though I had to lump to genus for some of them in order to be able to include them in analysis.

Table 1; some columns don’t line up. They were in the original; I have re-ordered the columns in this version. I will work with the editor to correct this.

Table 2; can you move the header to the same page as the table? Yes, this is just the way the review version came out

157 refers the reader to Figure 4-6, but where are they?

164 Table 3. Where is it? Found it, but a long ways away.

216 Fig 7. Where is it?

I expect figures and tables to follow the paragraph where they are first mentioned.

257 Table 4?

My mistake in preparing the file for the figures. Also, Tables 3 and 4 are long (Table 3 was in landscape orientation originally). I will work with the editor to place the figures more appropriately.

I like the NMDS plots Thank you

Figure 7. What do the widths indicate? The widths indicate the number of traps that contained a given number of specimens. The long thin parts of the plots at the lower end show that were few traps with few individuals, while thin parts at the top show that there were few traps with very many individuals.

Fig 7; why is the y axis shorter in 7C than the other two, when the scale is the same? I think it’s because at the grassland site there were not as many samples with low numbers of individuals as in the two forest habitats, that is, the “violins” were shorter.

Discussion

422-423 You give the authors for two of the three studies; why not all three.  My oversight - added the author names in the text

437 good conclusion

451 …under the tree canopy… modified canopy to tree canopy throughout

458 del comma after abundant  I deleted the comma

466 …no dominant species… “species” added

522 in some places you said % years, or 10 years, and no it is 15+? The different authors used different measures of “recovery.” In general, relative abundance is restored in 3 years or so, but species composition may take much longer to re-form. I change the text to “up to 15+ years” And the papers you cite are not from NM or even the southwest? (looks like 6 is; not sure about 37) Reference 6 is from Colorado and 37 is from arid Australia. There are not many papers from the southwestern US that look at arthropods. The region is under-collected in general, with or without disturbance. It’s one reason why I was fortunate to be able to work on this project. The species lists here are the first ones for the Caldera. More work has been done in Canadian forests.

528 Excellent conclusion

I would actually like to see Fig S1 in the paper. I have removed the precipitation figure and replaced it with the site map.

References

Excellent job with consistent format, but they expect this format: Aquat. Bot. 199963, 169–196.

That was the format in my original; I have re-formatted the entries.

Does the journal not require doi?  I have looked up doi’s for those references without them – could not find for all.

Reviewer 3 Report

Review of diversity-952091 titled “Responses of ground-dwelling spider (Arachnida: Araneae) communities to wildfire in three habitats in northern New Mexico, with notes on mites and harvestmen (Arachnida: Acari, Opiliones)”.

The manuscript presents the effect of wildfires on arachnid communities in three habitats, and how fast these communities recover. The manuscript is interesting and worth publishing in Diversity. The manuscript needs some more attention, but my concerns are not major:

The opening paragraph of the introduction starts with the goal of the study. In my opinion, the introduction lacks an opening paragraph or two, introducing the general topic of wildfires and their effect on biodiversity.

In results, you explain that wildfire affects arachnid communities significantly in all three habitats, and you mention the direction of change as similar, as well as how year and other parameters come into play. Could you rephrase this text so that the reader has a more vivid picture of what kind of changes wildfires cause on the community level?

You present results of spider abundances as number per trap. However, you claim that these relative abundance numbers are affected by the uneven trap numbers due to the loss of some samples in between the sampling period. Since there are no differences among sites within habitat, why would you claim that?

Lines 424-427: yes, in spider inventarisations there are several species that are caught with few specimen (singletons, doubletons), independent of collecting method. Thus, long term monitoring could also be conducted using more than one sampling method (pitfall traps).

From line 464 on: you first state that linyphiids are generally rare, and then list examples of these family being dominant in temperate communities around the world. In fact, linyphiids are the among the most diverse and abundant in most temperate habitats. Isn’t it likely that your findings reflect the sampling methodology, not actual species abundances? In this light, it makes sense to discuss changes and differences among sites/habitats, not so much species abundances per se?

Line 28: please include USA after New Mexico, to specify for non-US readers.

Lines 56-57: please rephrase question two. The goal was to how fast the burned areas recover (communities become increasingly more like the unburned areas), right?

Figure 3: it is impossible to read which error bars correspond to which group.

Figure 7: please mark significant differences.

Line 209: what and where ranged in abundance from 63.17 % to 69.68 %? Similarly, I’m confused in line 243.

I do not understand why Fig. 2 is important for this study?

Author Response

Diversity manuscript reviews – Sandra Brantley’s responses are in italics

Reviewer 3

The manuscript presents the effect of wildfires on arachnid communities in three habitats, and how fast these communities recover. The manuscript is interesting and worth publishing in Diversity. The manuscript needs some more attention, but my concerns are not major:

 The opening paragraph of the introduction starts with the goal of the study. In my opinion, the introduction lacks an opening paragraph or two, introducing the general topic of wildfires and their effect on biodiversity.

The literature on wildfire and diversity often focuses on tree/forest regeneration or replacement with herbaceous plants, grasses, or shrubs.

I have inserted this text: “In the southern Rocky Mountains several climate change factors are influencing the frequency and severity of fires in this region: trends towards larger fires, warmer maximum air temperatures between September and November, less precipitation between June and November, increased drought severity [2]. Results from a study of the years 1973-2012 in the western United States suggested trends of overall lengthening of the fire season from 37-117 days and mean burn time from 5 to 37 days [4]. Early snowmelt from the mountains also increased the likelihood of large summer wildfires [4]. In the Colorado Front Range of the Rocky Mountains Rother and Veblen [3] looked at stands of ponderosa pine from past fires to estimate what future climate changes could mean for tree establishment. They found that the severity of the burn was less important than the extent of the fire, making the distance to seed sources for new trees longer. More vulnerable stands of ponderosa pine were at their lower elevation level or were found on south-facing slopes. The effects of fires in forests are complex; season, tree species, and local habitat features (such as slope or aspect) are all important in affecting the outcome of wildfire.”

In results, you explain that wildfire affects arachnid communities significantly in all three habitats, and you mention the direction of change as similar, as well as how year and other parameters come into play. Could you rephrase this text so that the reader has a more vivid picture of what kind of changes wildfires cause on the community level?

I added a couple of sentences here.

Even with uneven sampling, PERMANOVA effects of wildfire and year were significant in all three habitats, even for the lightly burned MV grassland. The interaction between fire and year was significant for PP and MC, but not MV (Table 1). These effects were visualized with NMDS by treatment and year (Table 2, Figures 4-6). Besides losing tree canopy cover, on the ground the forest habitats lost litter cover, much of which was replaced with cover by grasses and herbaceous plants (Figure 3). The cover that had previously been there included pine needles, dead aspen leaves, and downed tree branches; a quite different microhabitat for small ground-dwelling arachnids.

 You present results of spider abundances as number per trap. However, you claim that these relative abundance numbers are affected by the uneven trap numbers due to the loss of some samples in between the sampling period. Since there are no differences among sites within habitat, why would you claim that? What were lost were 3 whole unburned sites; within-site trap loss was minimal for the remaining sites.

 Lines 424-427: yes, in spider inventarisations there are several species that are caught with few specimen (singletons, doubletons), independent of collecting method. Thus, long term monitoring could also be conducted using more than one sampling method (pitfall traps). I agree that long-term monitoring is the way to go. For this project, funding ended in 2015 for arthropods, although vegetation, soils, and at least some vertebrate animals are still being monitored. It is also difficult to get arthropods collected, sorted, and identified quickly if many taxa are involved. Expertise in many groups is being lost as well. Ideally, multiple sampling methods would have been used but that would also have increased the burden on the field crew at the Caldera. I was happy to get pitfall traps included.

 From line 464 on: you first state that linyphiids are generally rare, and then list examples of these family being dominant in temperate communities around the world. In fact, linyphiids are the among the most diverse and abundant in most temperate habitats. Isn’t it likely that your findings reflect the sampling methodology, not actual species abundances? In this light, it makes sense to discuss changes and differences among sites/habitats, not so much species abundances per se? I agree that the linyphiids are diverse and abundant at the family level. At the species level they can be collected in rather small numbers, especially in my region, which is semi-arid (even the forest enviroments are often dry). Linyphiids have not been looked at in much detail here and I am working to improve our knowledge and specimen base. The results did look at overall relative abundance, which was dominated by lycosids and gnaphosids (and harvestmen sometimes). Their numbers were often much higher than the linyphiids, but I wanted to find a way to include them as well, and grouping by genus allowed me to keep more linyphiid taxa in the analyses.

 Line 28: please include USA after New Mexico, to specify for non-US readers. I have added it on line 28, in the title, and in studies that took place in various US states.

 Lines 56-57: please rephrase question two. The goal was to how fast the burned areas recover (communities become increasingly more like the unburned areas), right? I have tried to clear up the question. The land managers at the Caldera wanted to know if 5 years is enough time for these arthropods to “recover” in abundance, so they can plan for future monitoring efforts. The managers are monitoring plants, animals, and abiotic conditions, and would like to conserve resources whenever possible. The problem with the comparisons is that the unburned areas were also changing every year, so they became a “moving target.” Year was important as a factor but not in a sequential or autocorrelated way. I ended up using the adonis pairwise comparisons for the years to show how or if the years differed. I also added Supplementary Material Figure S2, the scatterplots showing both years and treatment for the points. There is much variation, making it difficult to pick out any trend.

 Figure 3: it is impossible to read which error bars correspond to which group. I have remade the graphs so that the information is clearer. Editor: Please note that I cannot enter the letters b, c, and d for the panels for the figure 3.

Figure 7: please mark significant differences. I have removed Figure 7 and replaced it with Supplemental Material Figure S2.

 Line 209: what and where ranged in abundance from 63.17 % to 69.68 %? Similarly, I’m confused in line 243. I amended these sentences to clarify that the numbers are the percent species contribution to relative abundance for the given years.

 I do not understand why Fig. 2 is important for this study? I have removed the precipitation graphs and replaced them with the map of the study sites.

I have removed the precipitation Figure 2 and replaced it with the map of sampling sites, which becomes the new Figure 2. I have added this text in the methods section to replace the old figure: “In New Mexico most rainfall occurs during July-September (summer or monsoon pattern). At the Caldera from 2011-2015, rainfall ranged from 111-160mm per month in the summer, while amounts ranged from 0-50mm per month at other times of year.”

Reviewer 4 Report

This study is timely as large-scale wild fires more and more emerge as significant threat to biodiversity. It is therefore an extremely important study of high relevance.

The author must be congratulated on a fine manuscript and my comments are fairly minor. Most issues are addressed in comments in the attached manuscript, but summarised, I would say:

The introduction must be improved. Whilst the author mentions a number of relevant studies, we do not get to know what the core results of these studies are! In a well designed study, these results would form the base of hypotheses that will be tested. We do not get any of that and that lets the introduction down. This manuscript can be greatly improved by framing it more scientifically by testing clearly laid out hypotheses that will be tested with the design of the study, not just as 'I am just showing you what is changing or not'.

The methods are clearly laid out in most parts, but I have found some inconsistencies that need clarification (see text). I can follow the argument that the analysis is only done at the genus level, although it escapes me why not at least trying a species level analyses when species were actually identified. It seems almost as that has been done, the results are not as clear, so let's not talk about it? Due to the lack of species-level considerations, I rated 'scientific soundness' 'average'. Similarly, why not adding habitat as co-factor in the analyses? With the large dataset, I would still think it would provide interesting results. Just saying the capture probability of species is different, doesn't for me really justify this omission. What would be the alternative if one would want to compare habitats?

The results are well presented, although I found a few places, where discussion or interpretation sneaked in. I would take these sentences out here and move to the discussion.

The discussion overall is good, but as a result of the study design and overall approach (higher level analyses), some statements are too generalised, i.e. for Linyphiidae at the family level. My opinion is that any broader patterns should still be explained by species patterns, in particular if you have done the identifications at that level! In Linyphiidae/Erigone, it's done the other way, i.e. citing other studies and their general pattern and then being surprised when a single species doesn't follow it. That doesn't make sense.

Author Response

Diversity manuscript reviews – Sandra Brantley’s responses are in italics

Reviewer 4

Abstract

PERMANOVA mostly at the genus level with some higher taxon levels showed showed significant fire, year, and interaction effects.

Abundance was at or near unburned levels by 2014, but there were taxon changes in burned areas.

Introduction

Lines 50-53 Reviewer requested more info in this paragraph

I have added this text: “There are numerous papers on spider responses to prescribed fires, with and without the combined treatment of thinning [12-14], an increasing number on wildfire [6, 15-17], but understandably fewer with information on pre-fire conditions for the target species [18]. In different forests, spider abundance was either not affected by prescribed burns or returned to pre-fire levels within 3 years: Oregon, USA [12], in Swedish boreal forest [14], and in juniper-poplar steppe in Hungary [17]. Wildfires sometimes more strongly affected spider communities. In forest in Finland [15], spider assemblages were clearly different 3 years post-fire. In Canada spruce forest, a comparison of clear-cutting and wildfire [16] showed that spider responses to the two treatments were different and that wildfire (that is not catastrophic) could leave a more heterogeneous litter layer and thus had a less pronounced habitat effect than clear-cutting. In an Oregon, USA, grass/shrub steppe [18], study sites were in place before a wildfire occurred. There was no difference among sites before the fire, but afterwards community composition changed, although at the broader scale, richness and abundance did not. One study from Colorado, USA [6] looked at the effects of a 2002 catastrophic wildfire in pinyon-juniper woodland. Five-six years later, vegetation cover of grasses and annual plants had increased, litter decreased and bare ground increased, compared with nearby old-growth stands. None of the 32 spider species was in the top 7 indicators of burned or unburned sites, but abundance and richness had not returned to levels found in old-growth controls. Four spider species were positively associated with the burned areas and 16 were negatively associated with the burned areas. Given the variability in extent and severity of wildfire, it is difficult to compare it to prescribed fire, therefore additional studies from wildfire are needed. This is especially important for those areas that may be forested but occur in largely arid regions, as is the case for much of the southwestern United States.”

Line 95 pitfall length was changed to pitfall height

Lines 109-110 I have rewritten the abundance sentence: “Abundance as used here is a combination of actual abundance in the habitat and a measure of activity, since more active species are likely to be captured more often and so appear more abundant.”

Lines 111-112 why not use species- and genus-level analyses and compare?  I do have another dataset that is not so uneven that may be useful for such an analysis. It’s the data for the Thompson Ridge fire (that would use the data that couldn’t be included here. I think it can be set up rather like a before-after-control (BACI) design as well, because there are data from the control sites before they burned.

Line 125 I have added the species accumulation curves as a new Supplemental Material Figure S1.

Line 132 why not compare all three habitats in one study? I plead lack of resources to help me manage that. The Caldera program was most interested in changes over time within each habitat. But see my answer to Lines 111-112 above.

Line 133 say activity abundance rather than abundance I modified the text

Results

Lines 150-152 move to methods I removed from the Results

Line 165 Table 1 Permanova results for I addedarachnid genera (and some higher taxa)”

Line 188 Table 2 added burned/unburned – added text

Lines 212-213 Changed to abundance was affected

Lines 227-229 move to discussion: The gnaphosid genus Micaria is well-known for ant mimicry, both in associating with ants, and preying on them (Platnick and Shadab 1988). Knudsen [32] reported increased ant abundance in PP in burned sites after Spring 2012, which may have influence Micaria activity abundance as well. I removed this from the Results

Line 244 expected  I have removed the word

Lines 260-261 I removed the Callobius sentence

Line 265 first sentence should be in intro/methods? I noted the extent of the burn in the Introduction and mentioned it again at the start of each habitat section as a reminder.

Lines 270-271 move to discussion: the genus being found more often in high-elevation forests in the southern part of its range, which includes New Mexico. I have moved the sentence out of Results

Lines 276-277 move to discussion “Some philodromids…(Figure 3).  I have moved this to the Discussion.

Lines 286-287 move to discussion: “the presence of Micaria…[32]”. I have moved this to the Discussion.

Line 289 move to discussion: “which is common in grassy areas.” I have removed the text.

Line 294 add burned/unburned added

Discussion

Line 420 By analyzing separately… I removed the sentence because you are right that I did not test the habitats together to make the comparison.

Lines 421-423 about abundance numbers back up in 3 years – That is in the abstract as you suggested.

Line 442 small litter-dwellers – need examples. .”…were the small litter-dwellers, which included such genera as, Helophora, Sisicottus, and Wubana in the Linyphiidae, and Neoantistea in the Hahniidae.”

Lines 457-463 notes on Pardosa species – I moved this section to the General Results

Line 466  “no dominant because abundances were low” I left a word out – “no dominant species.” I responded to one of the other reviewers that while the linyphiids were indeed abundant at the family level, at the species level it may be more difficult to collect enough for analysis. This may be a problem in drier habitats.

Lines 468-472 comment too generalized at the family level for Linyphiidae. True – what I meant to say and left out was that these were the linyphiid genera that were indicators of unburned habitat. I will clarify the text.

Lines 473-474 the high numbers of Erigone dentosa. I removed the first part of the sentence, so it now begins with “Hundreds of E. dentosa were collected…”

Line 479 no surprise that most linyphiids were in unburned areas because they require vegetation for webs. I would add they need litter cover too for those living at or near ground-level. The vegetation cover increased in the burned areas – it may just have been the wrong structure or didn’t produce the right litter, like leaves.

Lines 494-495 Mesostigmata mites taking longer (compared to erythraeids) to return to “normal” abundance levels both at the Caldera and in Swedish forests. Why might that be? I don’t know. Some ideas are a slow reproductive rate, slow growth rate, or lack of a prey base that isn’t restored for several years. Yes, just found in the Krantz and Walter acarology text that female mesostigs may spend a lot of time and energy finding and placing their eggs in optimal locations – such as attaching eggs to plant rootlets so the larvae hatch in a place where symphylan prey occur!

Lines 505-506 I recommend analyses at more than one taxon level and then didn’t do them. Right. I’ll remove that line.

Lines 506-508 spiders may be prey generalists but definitely are specialists in terms of microhabitat. I don’t disagree with you on this and will clarify my text.

Line 511 arid Australia takes a longer time to recover than our forested areas. I will clarify the text – I wanted to make two points from the Langlands paper 1) even for these small-bodied animals with relatively short lifespans, they may need more time to return to previous abundance levels than we think, and 2) the paths to “recovery” can vary by habitat or even site. The spiders in the two forests in this study had different trajectories over the five years (the NMDS’s are different and the scatterplots are different in S2).

Line 524 loss of rarer species I will change this to loss of habitat specialists